# BUB-1 promotes amphitelic chromosome biorientation via multiple activities at the kinetochore

**Frances Edwards[1], Gilliane Maton[1], Nelly Gareil[1], Julie C Canman[2], Julien Dumont[1]\***

[1]Institut Jacques Monod, CNRS, UMR 7592, University Paris Diderot, Sorbonne Paris Cité, Paris, France; [2]Department of Pathology and Cell Biology, Columbia University, New York, United States

**Abstract** Accurate chromosome segregation relies on bioriented amphitelic attachments of chromosomes to microtubules of the mitotic spindle, in which sister chromatids are connected to opposite spindle poles. BUB-1 is a protein of the Spindle Assembly Checkpoint (SAC) that coordinates chromosome attachment with anaphase onset. BUB-1 is also required for accurate sister chromatid segregation independently of its SAC function, but the underlying mechanism remains unclear. Here we show that, in *Caenorhabditis elegans* embryos, BUB-1 accelerates the establishment of non-merotelic end-on kinetochore-microtubule attachments by recruiting the RZZ complex and its downstream partner dynein-dynactin at the kinetochore. In parallel, BUB-1 limits attachment maturation by the SKA complex. This activity opposes kinetochore-microtubule attachment stabilisation promoted by CLS-2[CLASP]-dependent kinetochore-microtubule assembly. BUB-1 is therefore a SAC component that coordinates the function of multiple downstream kinetochore-associated proteins to ensure accurate chromosome segregation.
DOI: https://doi.org/10.7554/eLife.40690.001

**\*For correspondence:**
julien.dumont@ijm.fr

**Competing interests:** The authors declare that no competing interests exist.

## Introduction

During mitosis, the microtubule-based spindle segregates sister chromatids by attaching to the macromolecular kinetochores assembled on centromeres. Accurate segregation of chromosomes requires their biorientation by amphitelic attachments connecting each sister chromatid to microtubules emanating from opposite spindle poles. In metazoans, two main kinetochore microtubule-binding modules ensure efficient interaction with spindle microtubules: the NDC80 complex and the Rod-Zw10-Zwilch (RZZ)-Spindly kinetochore module that recruits dynein-dynactin motors to kinetochores (*Gassmann et al., 2008a*; *Griffis et al., 2007*; *Scaërou et al., 1999*; *Starr et al., 1998*; *Williams et al., 2003*). The NDC80 complex interacts with microtubule plus-ends to establish load-bearing end-coupled connections between kinetochores and the mitotic spindle (*Cheeseman et al., 2006*; *DeLuca et al., 2005*; *McCleland et al., 2003*; *Wigge and Kilmartin, 2001*). Kinetochore-localized dynein mediates initial lateral capture of microtubules to ensure correct kinetochore orientation (*Gassmann et al., 2008a*). This initial kinetochore orientation limits merotelic connections, where individual kinetochores are attached to both spindle poles, and accelerates formation of NDC80-mediated end-coupled attachments (*Rieder and Alexander, 1990*; *Rieder and Salmon, 1998*; *Varma et al., 2008*). The transition from initial lateral to load-bearing end-coupled attachments is coordinated by the RZZ complex itself, which inhibits NDC80 binding to microtubules until dynein has properly oriented kinetochores (*Cheerambathur et al., 2013*). Once properly established, load-bearing connections are subsequently reinforced by the SKA complex, which interacts with NDC80 and kinetochore microtubules (*Auckland et al., 2017*; *Cheerambathur et al., 2017*;

*Gaitanos et al., 2009*; *Helgeson et al., 2018*; *Janczyk et al., 2017*; *Schmidt et al., 2012*; *Theis et al., 2009*). During chromosome alignment, several mechanisms promote amphitelic load-bearing attachments (reviewed in (*Auckland and McAinsh, 2015*; *Maiato et al., 2017*). Chromosome biorientation through amphitelic attachment is selectively self-stabilised by the tension generated on kinetochores. This tension prevents Aurora B kinase-dependent destabilisation of attachments (reviewed in *Lampson and Grishchuk, 2017*), and/or promotes TOG (Tumour Over-expressed Gene) domain protein-dependent stabilisation of attachments (*Miller et al., 2016*). Merotely is also limited by initially high kinetochore microtubule dynamics, which maintain a high turnover of kinetochore-microtubule attachments until biorientation is achieved (*Bakhoum et al., 2009a*; *Bakhoum et al., 2009b*; *Lampson and Grishchuk, 2017*).

An additional key player in the process of chromosome alignment is the kinetochore BUB1 kinase that was originally identified as a component of the Spindle Assembly Checkpoint (SAC), a mechanism that delays anaphase onset until all kinetochores are properly connected to the spindle (reviewed in *Joglekar, 2016*). BUB1 is also directly involved in establishing proper chromosome attachments to the mitotic spindle. Indeed, independently of the SAC, BUB1 was shown to be essential to preserve ploidy throughout mitosis in budding and fission yeasts (*Bernard et al., 1998*; *Vanoosthuyse et al., 2004*; *Warren et al., 2002*), to promote chromosome alignment and accurate segregation essential for embryonic development in *C. elegans* oocytes and embryos (*Essex et al., 2009*; *Laband et al., 2017*), and to ensure efficient chromosome congression and segregation and prevent aneuploidy in human tissue cultured cells (*Johnson et al., 2004*; *Klebig et al., 2009*; *Meraldi and Sorger, 2005*). However, the molecular mechanism(s) behind these non-SAC functions of BUB1, and how BUB1 influences chromosome attachments to the mitotic spindle remain unclear.

BUB1 mediates the recruitment of several downstream proteins involved in accurate chromosome segregation. BUB1 phosphorylation of histone H2A (*Kawashima et al., 2010*; *Ricke et al., 2012*) and subsequent recruitment of the centromeric protein Shugoshin (*Fernius and Hardwick, 2007*) target protein phosphatase 2A (PP2A) and Aurora B kinase to the inner centromere (*Kawashima et al., 2010*; *Yamagishi et al., 2010*). In parallel, BUB1 interacts with BUBR1, which leads to kinetochore recruitment of PP2A (*Zhang et al., 2016*). Aurora B and PP2A are in turn thought to promote accurate chromosome segregation via their kinetochore regulatory functions (reviewed in *Saurin, 2018*). However, several studies suggest that BUB1 can regulate kinetochore-microtubule attachment independently of this pathway (*Johnson et al., 2004*; *Meraldi and Sorger, 2005*; *Perera and Taylor, 2010*; *Raaijmakers et al., 2018*; *Windecker et al., 2009*). A second function of BUB1 that could account for its role in regulating kinetochore-microtubule attachments is the recruitment of the kinetochore protein CENP-F, via its kinase domain but independently of kinase activity (*Berto et al., 2018*; *Ciossani et al., 2018*; *Encalada et al., 2005*; *Johnson et al., 2004*). However, the role of CENP-F during chromosome segregation remains controversial (*Bomont et al., 2005*; *Cheeseman et al., 2005*; *Holt et al., 2005*; *McKinley and Cheeseman, 2017*; *Pfaltzgraff et al., 2016*). Alternatively, as BUB1 contributes to recruiting the RZZ complex to kinetochores in human cells (*Caldas et al., 2015*; *Zhang et al., 2015*), kinetochore-localized dynein-dynactin may be required downstream of BUB1 for establishing amphitelic attachments. Yet, the loss of BUB1-dependent kinetochore targeting of the RZZ complex is not sufficient to account for the lethality of SAC-deficient haploid (HAP1) cells upon BUB1 depletion (*Raaijmakers et al., 2018*). Altogether, these observations suggest that BUB1 contributes to chromosome biorientation and alignment through one or several additional downstream activities that remain to be identified.

Here, by analysing BUB-1 during mitosis in the *C. elegans* one-cell embryo or zygotes, we show that it has multiple functions essential for proper chromosome biorientation. Coordination of these activities by BUB-1 is required to prevent merotely and embryonic lethality. We therefore propose that BUB-1 is a key regulator of kinetochore-microtubule interactions that ensures accurate chromosome biorientation and segregation.

## Results

### BUB-1 inhibits chromosome biorientation

To study BUB-1 function in kinetochore-microtubule attachments, we performed live imaging of control or BUB-1-depleted *C. elegans* one-cell embryos. To monitor chromosome segregation, we used

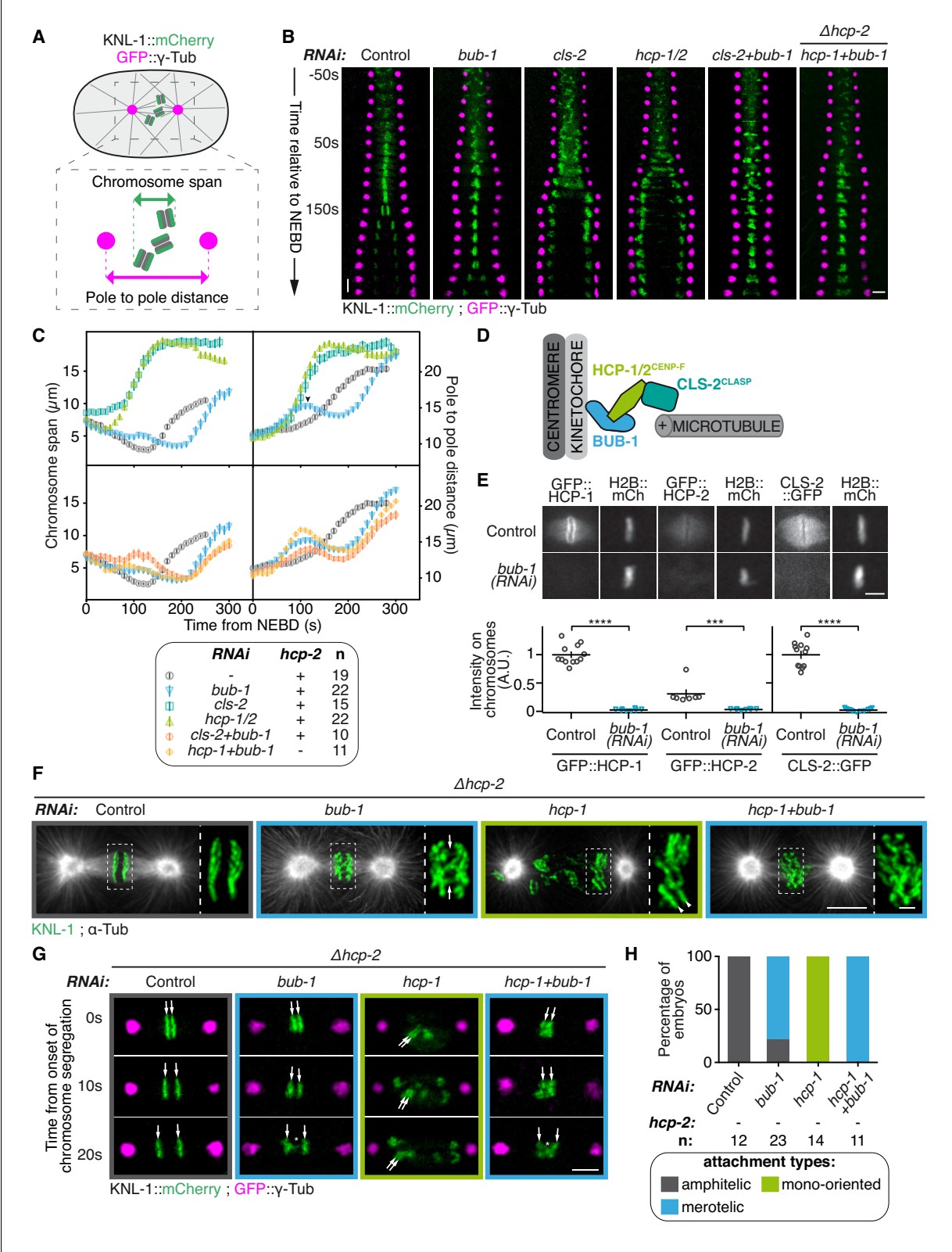

**Figure 1.** BUB-1 inhibits chromosome biorientation. (A) Assay for kinetochore-microtubule attachment formation and chromosome congression. GFP::γ-Tub is used to measure the pole to pole distance, and KNL-1::mCherry is used to measure the chromosome span in the spindle pole axis. (B) Kymographs generated from embryos expressing GFP::γ-Tub and KNL-1::mCherry, for the different indicated conditions. Horizontal scale bar, 5 μm; Vertical scale bar, 20 s. (C) Chromosome span and pole to pole distance as functions of time after NEBD for the indicated conditions. Top right corner:

*Figure 1 continued on next page*

*Figure 1 continued*

Arrowhead, spindle pole bump following BUB-1 depletion. (D) Schematics of BUB-1 at kinetochores recruiting its downstream partners HCP-1/2[CENP-F] and CLS-2[CLASP]. (E) Top: Representative images from time-lapse movies showing BUB-1 dependent localisations of GFP::HCP-1[CENP-F], GFP::HCP-2[CENP-F] and CLS-2[CLASP]::GFP on chromosomes (H2B::mCherry), at metaphase. Bottom: Quantification of the GFP signal on chromosomes at metaphase. Mann Whitney tests were used to determine significance (GFP::HCP-1 p < 0.0001, GFP::HCP-2 p = 0.0003, CLS-2::GFP p < 0.0001). Scale bar, 5 µm. (F) Immunofluorescent staining of kinetochores (KNL-1) and microtubules (DM1α) in Δhcp-2 zygotes at metaphase in the indicated conditions. Scale bar, 5 µm. Magnifications of the kinetochore region (highlighted by a dashed rectangle) are shown on the right of each panel. Arrows point to bent merotelic kinetochores in the BUB-1-depleted zygote. Arrowheads show a mono-oriented chromosome in the HCP-1[CENP-F]-depleted zygote. Scale bar, 1 µm. (G) Representative images of kinetochores (KNL-1::mCherry, green) and spindle poles (GFP::γ-Tub, magenta), at different times from the onset of chromosome segregation, for the indicated conditions. White arrows point towards sister kinetochores. White asterisks indicate the presence of kinetochore stretches. Scale bar, 5 µm. (H) Quantification of the percentage of embryos with chromosomes engaged in amphitelic, merotelic and mono-oriented attachments, in the indicated conditions. Error bars represent the SEM.

DOI: https://doi.org/10.7554/eLife.40690.002

The following source data and figure supplements are available for figure 1:

**Source data 1.** Chromosome span and pole to pole distance as functions of time after NEBD for the indicated conditions.
DOI: https://doi.org/10.7554/eLife.40690.008

**Source data 2.** GFP::HCP-1, GFP::HCP-2, and CLS-2::GFP signals on chromosomes at metaphase.
DOI: https://doi.org/10.7554/eLife.40690.009

**Source data 3.** Percentage of embryos with chromosomes engaged in amphitelic, merotelic and mono-oriented attachments, in the indicated conditions.
DOI: https://doi.org/10.7554/eLife.40690.010

**Figure supplement 1.** HCP-1/2[CENP-F] and CLS-2[CLASP] downstream of BUB-1 prevent premature chromosome segregation.
DOI: https://doi.org/10.7554/eLife.40690.003

**Figure supplement 1—source data 1.** GFP::HCP-1, GFP::HCP-2, and CLS-2::GFP signals on chromosomes over time.
DOI: https://doi.org/10.7554/eLife.40690.004

**Figure supplement 1—source data 2.** Quantifications of the integrated anaphase sensor GFP signal measured on chromosomes over time.
DOI: https://doi.org/10.7554/eLife.40690.005

**Figure supplement 2.** Characterization of the Δhcp-1 and Δhcp-2 alleles.
DOI: https://doi.org/10.7554/eLife.40690.006

**Figure supplement 2—source data 1.** Chromosome span and pole to pole distance as functions of time after NEBD for the indicated conditions.
DOI: https://doi.org/10.7554/eLife.40690.007

a strain expressing KNL-1::mCherry to label kinetochores and GFP::γ-Tubulin to track spindle poles (*Figure 1A*; *Video 1*). Kymographs allowed visualising the behaviour of kinetochores and spindle poles over time (*Figure 1B*). Kinetochores are holocentric in the *C. elegans* nematode and form along the length of sister chromatids (*Dernburg, 2001*), thus chromosome congression, alignment and segregation were quantified by measuring the space occupied by kinetochores in the spindle pole axis (chromosome span) over time (*Figure 1A,C*). In *C. elegans* zygotes, spindle pole separation is primarily driven by cortical pulling forces transmitted through astral microtubules (*Grill et al., 2001*). Cortical forces are then opposed by load-bearing connections between bioriented chromosomes and the two poles of the mitotic spindle. We therefore also measured spindle elongation (pole to pole distance) over time as an indirect read-out of these connections (*Figure 1A,C*) (*Desai et al., 2003*).

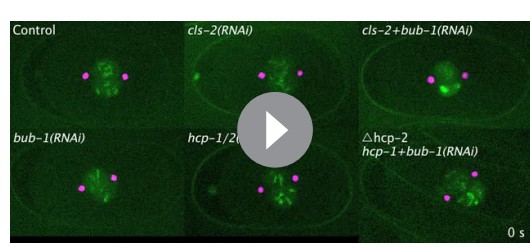

**Video 1.** One-cell *C. elegans* embryos in the indicated conditions. 10 s per frame. Magenta, γ-Tubulin::GFP (spindle poles); Green, KNL-1::mCherry (kinetochores).
DOI: https://doi.org/10.7554/eLife.40690.011

In control zygotes, progressive chromosome congression led to the formation of a tight metaphase plate (*Figure 1B,C*; *Video 1*). This correlated with slow spindle pole separation from 80 s after nuclear envelope breakdown (NEBD) until anaphase onset (140 s after NEBD), and was followed by sister chromatid segregation concomitant with fast spindle pole separation (*Figure 1B,C*). In BUB-1-depleted zygotes, anaphase onset was delayed compared to controls (around 200 s after NEBD, *Figure 1B,C*), in agreement with BUB-1 promoting anaphase onset through Cdc20 activation (*Kim et al., 2017*; *Kim et al., 2015*; *Yang et al., 2015*). In

these embryos, a chromosome congression pause was visible in the kymographs, as evidenced by a constant chromosome span that coincided with a previously described 'bump' in the pole separation profile (*Kim et al., 2015*). This spindle pole bump has been shown to reflect delayed load-bearing attachments (*Kim et al., 2015*), which are eventually established to allow metaphase plate formation (*Figure 1B,C*; *Video 1*). Sister chromatids were connected to opposite spindle poles, but these attachments were not amphitelic. Instead, bent kinetochores during metaphase and kinetochore stretches during anaphase were visible in BUB-1-depleted zygotes, which in holocentric embryos indicate merotelic connections of individual kinetochores to both spindle poles (*Figure 1F–H*) (*Essex et al., 2009*; *Maton et al., 2015*). BUB-1 depletion therefore delays load-bearing attachments, leading to a lag in chromosome congression, followed by the establishment of merotelic attachments to both spindle poles.

We next sought to test which proteins downstream of BUB-1 also affect chromosome attachment or segregation. BUB-1 is required for kinetochore recruitment of the two redundant CENP-F orthologs HCP-1 and 2, which in turn recruit CLS-2$^{CLASP}$ (*Figure 1D*) (*Cheeseman et al., 2005*; *Maton et al., 2015*). CLASP-family proteins are key regulators of kinetochore microtubule dynamics, which promote kinetochore microtubule growth while limiting their stability (*Cheeseman et al., 2005*; *Lacroix et al., 2018*; *Maffini et al., 2009*; *Maiato et al., 2005*; *Maiato et al., 2003a*; *Pereira et al., 2006*). We imaged *C. elegans* zygotes expressing endogenous GFP-tagged HCP-1$^{CENP-F}$ or HCP-2$^{CENP-F}$, or a GFP-tagged CLS-2$^{CLASP}$ transgene, and confirmed their absence from kinetochores upon BUB-1 depletion (*Figure 1E*; *Figure 1—figure supplement 1A,B*). In line with previous observations, following depletion of HCP-1/2$^{CENP-F}$ or CLS-2$^{CLASP}$, chromosomes did not congress, and chromosomes and spindle poles separated prematurely 40 s after NEBD concomitant with sister chromatid co-segregation towards the same spindle pole (*Figure 1B–H*; *Video 1*) (*Cheeseman et al., 2005*). This early pole and chromosome separation was not caused by premature anaphase onset, as the fluorescent signal of a Separase activation sensor remained on chromosomes well after they completed separation in absence of HCP-1/2$^{CENP-F}$ or CLS-2$^{CLASP}$ (*Figure 1—figure supplement 1C–E*) (*Kim et al., 2015*). Instead, a vast majority of chromosomes remained mono-oriented in HCP-1/2$^{CENP-F}$- or CLS-2$^{CLASP}$-depleted zygotes (*Figure 1F–H*) (*Cheeseman et al., 2005*). This indicates that the sister chromatid co-segregation phenotype resulting from depletion of HCP-1/2$^{CENP-F}$ or CLS-2$^{CLASP}$ is due to improperly mono-oriented chromosomes rather than premature mitotic exit. Therefore, while being epistatically related at the kinetochore, the depletion of BUB1 and its downstream partners, HCP-1/2$^{CENP-F}$ or CLS-2$^{CLASP}$, have very different effects on chromosome segregation: BUB-1 depletion results in delayed chromosome congression and bioriented yet merotelic attachments, while HCP-1/2$^{CENP-F}$ or CLS-2$^{CLASP}$ depletion induces mono-orientation and sister chromatid co-segregation.

We next considered two non-exclusive hypotheses to account for this apparent discrepancy. First, HCP-1/2$^{CENP-F}$ and CLS-2$^{CLASP}$ could play an essential function outside of the kinetochore and independently of BUB-1. Second, BUB-1 could be causing the phenotype observed upon HCP-1/2$^{CENP-F}$ or CLS-2$^{CLASP}$ depletion. To test the latter hypothesis, chromosome segregation was analysed in zygotes simultaneously depleted of BUB-1 and CLS-2$^{CLASP}$. Strikingly BUB-1 depletion restored biorientation and rescued the sister chromatid segregation failure typical of CLS-2$^{CLASP}$ depletion (*Figure 1B,C*; *Video 1*). We also observed similar sister chromatid segregation rescue by co-depleting BUB-1 and HCP-1$^{CENP-F}$ or HCP-2$^{CENP-F}$, in CRIPSR-cas9 generated *hcp-2 (Δhcp-2)* (*Figure 1B,C*; *Figure 1—figure supplement 2A–C*) and *hcp-1 (Δhcp-1*; *Figure 1—figure supplement 2D–E*) deletion mutants respectively. BUB-1 and HCP-1$^{CENP-F}$ (or HCP-2$^{CENP-F}$) co-depleted embryos displayed a 'BUB-1-like' phenotype with delayed congression (*Figure 1B,C*) and merotelic attachments to both spindle poles evidenced by bent kinetochores during metaphase and kinetochore stretches during anaphase (*Figure 1F–H*). We confirmed by immunofluorescence the absence of BUB-1 and HCP-1$^{CENP-F}$ after RNAi-treatment in the Δ*hcp-2* strain (*Figure 1—figure supplement 2F*). Together, these results suggest that BUB-1 prevents chromosome biorientation, and that this activity is revealed in absence of HCP-1/2$^{CENP-F}$ or CLS-2$^{CLASP}$.

## Biorientation inhibition requires BUB-1 localisation at the kinetochore

We next tested whether BUB-1 needs to be localised at the kinetochore to exert its inhibitory effects in absence of HCP-1/2$^{CENP-F}$ and/or CLS-2$^{CLASP}$. BUB-1 recruitment to kinetochores depends on phosphorylated MELT repeats located in the N-terminal part of the kinetochore scaffolding protein

KNL-1 (*Vleugel et al., 2013*). To abolish BUB-1 recruitment to kinetochores, we used a *C. elegans* strain expressing a truncation mutant of KNL-1 (KNL-1$^{\Delta 85-505}$), which lacks all MELT repeats (*Figure 2A*; *Figure 2—figure supplement 1A*) (*Moyle et al., 2014*). We then introduced KNL-1$^{WT}$ and KNL-1$^{\Delta 85-505}$ RNAi-resistant transgenes in the $\Delta hcp-2$ mutant background. As expected, depleting HCP-1$^{CENP-F}$ along with endogenous KNL-1 in absence of HCP-2 but in presence of the RNAi-resistant KNL-1$^{WT}$ transgene led to the same phenotype as in absence of HCP-1/2$^{CENP-F}$ only, characterised by sister chromatid mono-orientation followed by co-segregation (*Figure 2B–D*; *Figure 2—figure supplement 1B*; *Video 2*). In contrast, in the presence of KNL-1$^{\Delta 85-505}$, and therefore in the absence of BUB-1 at kinetochores (*Figure 2—figure supplement 1A*), co-depleted zygotes displayed a 'BUB-1-like' phenotype with delayed congression and bioriented merotelic attachments (*Figure 2B–D*; *Figure 2—figure supplement 1B*; *Video 2*). Together, these results show that BUB-1 inhibitory activity on biorientation depends on its kinetochore localisation.

## BUB-1 kinase domain inhibits biorientation independently of its kinase activity

To determine if BUB-1 kinase activity is required for inhibiting chromosome biorientation, we used three different RNAi-resistant mutant transgenes that affect the kinase domain and/or kinase activity of BUB-1 (*Figure 3A*): (1) a full truncation of the kinase domain (BUB-1$^{\Delta KD}$), (2) a dual point mutant that destabilizes the kinase domain and prevents its interaction with the SAC component MDF-1$^{Mad1}$ (BUB-1$^{K718R\ ;D847N}$), and (3) a point mutant with defective kinase activity by mutation of the catalytic aspartate in the 'HxD"motif (BUB-1$^{D814N}$) (*Kang et al., 2008*; *Moyle et al., 2014*). In line with previous findings on BUB-1 function in the regulation of anaphase onset, BUB-1$^{\Delta KD}$ and BUB-1$^{K718R\ ;D847N}$, but not BUB-1$^{D814N}$, led to delayed anaphase onset (*Figure 3—figure supplement 1A,B*) (*Kim et al., 2015*; *Yang et al., 2015*). Importantly, all of these mCherry-tagged mutants localised to kinetochores in absence of endogenous BUB-1, allowing us to test their influence on kinetochore-microtubule attachments (*Figure 3—figure supplement 1C*). In absence of endogenous BUB-1, all three mutant transgenes induced only mild variations in the chromosome span and pole separation profiles compared to control embryos, indicating that congression and load-bearing attachment are not drastically altered when BUB-1 kinase activity is impaired (*Figure 3—figure supplement 1A,B*; *Video 3*). Additionally, sister kinetochore alignment and segregation was similar to in control embryos, with only occasional merotelic attachments in the BUB-1$^{\Delta KD}$ mutant evidenced by anaphase kinetochore stretches (*Figure 3—figure supplement 1D,E*). Interestingly, BUB-1$^{\Delta KD}$ prevented HCP-1$^{CENP-F}$ kinetochore localization (*Figure 3—figure supplement 1C*) suggesting that BUB1 recruitment of CENP-F via its kinase domain is conserved from humans to *C. elegans* (*Ciossani et al., 2018*; *Raaijmakers et al., 2018*). Furthermore, the presence of mainly bioriented chromosomes with amphitelic attachments in this mutant that lacks kinetochore HCP-1/2$^{CENP-F}$, suggests that BUB-1's inhibitory role during biorientation is dependent on the kinase domain.

To confirm the kinase domain is essential, we analysed BUB-1 mutants in absence of endogenous BUB-1 and HCP-1/2$^{CENP-F}$ (*Figure 3B,C*). BUB-1$^{WT}$ led to premature pole and chromosome separation (*Figure 3B,C*) and the establishment of mono-oriented attachments (*Figure 3D*; *Figure 3—figure supplement 1E*), as expected in absence of HCP-1/2$^{CENP-F}$. In contrast, the phenotype was attenuated with BUB-1$^{\Delta KD}$ and BUB-1$^{K718R\ ;D847N}$: chromosome congression was delayed but occurred successfully (*Figure 3B,C*; *Video 3*), and chromosome biorientation with merotelic attachments was established (*Figure 3D*; *Figure 3—figure supplement 1E*). The appearance of a large proportion of merotelic attachments only after depletion of HCP-1/2$^{CENP-F}$ in BUB-1$^{\Delta KD}$ (which normally lacks kinetochore-associated HCP-1/2$^{CENP-F}$) suggests that HCP-1/2$^{CENP-F}$ promote non-merotelic attachments independently of its kinetochore localization (*Figure 3D* and *Figure 3—figure supplement 1D*). Furthermore, these results confirm that the integrity of the kinase domain is required for BUB-1 inhibition of chromosome congression and biorientation. In contrast, the HCP-1/2$^{CENP-F}$ loss-of-function phenotype was not rescued by BUB-1$^{D814N}$, which specifically disrupts the kinase activity (*Figure 3B–D*; *Figure 3—figure supplement 1E*) (*Moyle et al., 2014*). Altogether, these results demonstrate that BUB-1's inhibitory role on biorientation depends on its kinase domain, but not its kinase activity.

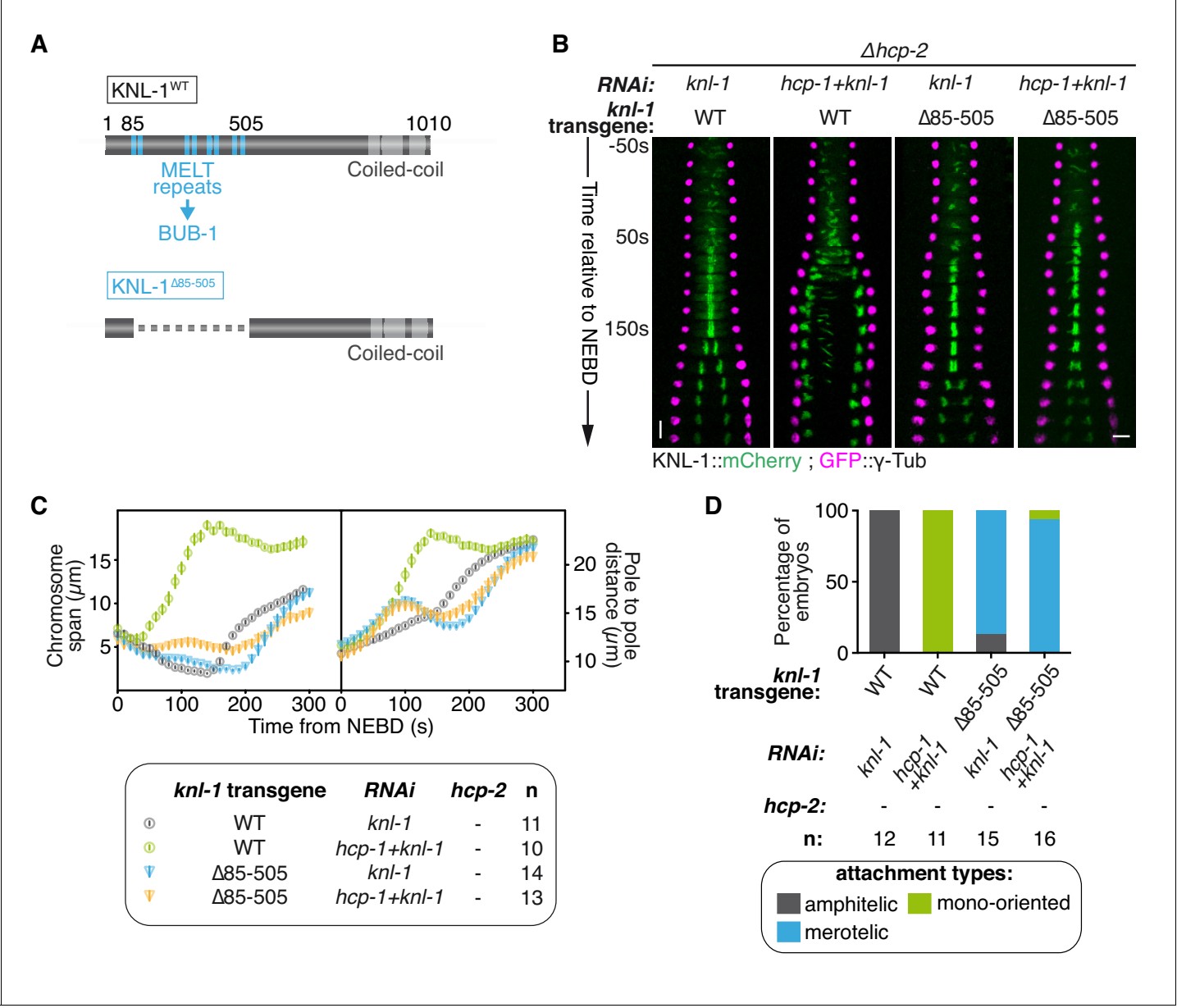

**Figure 2.** Biorientation inhibition requires BUB-1 localisation at the kinetochore. (**A**) Schematics of WT KNL-1 and of the Δ85–505 mutant that leads to loss of BUB-1 from kinetochores. (**B**) Kymographs generated from embryos expressing GFP::γ-Tub and KNL-1::mCherry, for the indicated conditions. (**C**) Chromosome span and pole to pole distance as functions of time after NEBD for the indicated conditions. (**D**) Quantification of the percentage of embryos with chromosomes engaged in amphitelic, merotelic and mono-oriented attachments in the indicated conditions. Error bars represent the SEM. Horizontal scale bar, 5 µm; Vertical scale bar, 20 s.

DOI: https://doi.org/10.7554/eLife.40690.012

The following source data and figure supplements are available for figure 2:

**Source data 1.** Chromosome span and pole to pole distance as functions of time after NEBD for the indicated conditions.
DOI: https://doi.org/10.7554/eLife.40690.015

**Source data 2.** Percentage of embryos with chromosomes engaged in amphitelic, merotelic and mono-oriented attachments, in the indicated conditions.
DOI: https://doi.org/10.7554/eLife.40690.016

**Figure supplement 1.** Biorientation inhibition requires BUB-1 localization at the kinetochore.
DOI: https://doi.org/10.7554/eLife.40690.013

**Figure supplement 1—source data 1.** BUB-1::GFP signal measured on kinetochores at metaphase in the indicated conditions.
DOI: https://doi.org/10.7554/eLife.40690.014

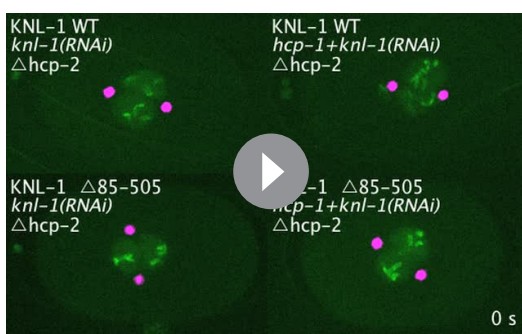

**Video 2.** One-cell *C. elegans* embryos in the indicated conditions. 10 s per frame. Magenta, γ-Tubulin::GFP (spindle poles); Green, KNL-1::mCherry (kinetochores).
DOI: https://doi.org/10.7554/eLife.40690.017

## BUB-1 accelerates non-merotelic end-on attachments via dynein, independently of the inhibitory effect on biorientation

We next sought to identify the mechanism by which kinetochore-localized BUB-1 prevents the establishment of chromosome biorientation in absence of HCP-1/2$^{CENP-F}$ or CLS-2$^{CLASP}$. In mammals, BUB1 contributes to recruiting the RZZ complex, and thus probably also dynein-dynactin, to kinetochores (*Caldas et al., 2015*; *Zhang et al., 2015*). Kinetochore-localized RZZ and dynein-dynactin could prevent chromosome congression by accelerating end-on attachments to short non-dynamic microtubules induced by the absence of HCP-1/2$^{CENP-F}$ or CLS-2$^{CLASP}$ (*Figure 4A*). This would promote mono-oriented connections and sister chromatid co-segregation.

In contrast, in absence of BUB-1, and thus without kinetochore-localized RZZ and dynein-dynactin, kinetochores are not properly oriented toward spindle poles (*Gassmann et al., 2008a*). In absence of HCP-1/2$^{CENP-F}$ or CLS-2$^{CLASP}$, this would result in largely merotelic connections to short non-dynamic microtubules, and could provide resistance to cortical traction forces thus restoring chromosome biorientation (*Figure 4A*).

To test this hypothesis, we first analysed the recruitment of RZZ and dynein-dynactin to kinetochores in BUB-1-depleted zygotes. Consistent with previous results in human cells, BUB-1 depletion led to a drastic reduction of the GFP-tagged RZZ complex component ZWL-1$^{Zwilch}$ at kinetochores (*Figure 4B*; *Figure 4—figure supplement 1A,B*). This result is also in line with the significant reduction in kinetochore-localized GFP::CZW-1$^{ZW10}$ that was previously observed in the KNL-1$^{Δ85-505}$ mutant, which lacks kinetochore-localized BUB-1 (*Maton et al., 2015*). Reduction in RZZ kinetochore targeting coincided with the complete loss of the GFP-tagged dynactin subunit DNC-2$^{p50}$ from kinetochores, which instead localized diffusely on chromosomes and in the spindle region (*Figure 4B*; *Figure 4—figure supplement 1C,D*). Therefore BUB-1 is required for dynein-dynactin recruitment to kinetochores in *C. elegans* zygotes, and the residual pool of RZZ complex that localises to kinetochores independently of BUB-1 does not contribute to dynein-dynactin recruitment. Importantly, both ZWL-1$^{Zwilch}$ and DNC-2$^{p50}$ were present at kinetochores in absence of HCP-1/2 $^{CENP-F}$ (*Figure 4B*; *Figure 4—figure supplement 1A–D*).

Next, we tested if RZZ and dynein-dynactin participate in the BUB-1-dependent inhibition of chromosome biorientation in absence of HCP-1$^{CENP-F}$. For this, we co-depleted ZWL-1$^{Zwilch}$ with HCP-1$^{CENP-F}$ in the Δ*hcp-2* strain. This led to a slight improvement in chromosome congression, and chromosome segregation occurred later and more slowly than in absence of HCP-1/2$^{CENP-F}$ alone (*Figure 4C,D*; *Video 4*). This result suggests that RZZ and dynein promote rapid end-on attachments and early co-segregation of sister chromatids in absence of HCP-1/2$^{CENP-F}$. However, chromosome biorientation was not restored and sister chromatid still co-segregated in this condition (*Figure 4C–E*; *Figure 4—figure supplement 1E*; *Video 4*). Further evidence confirming the independence of BUB-1 activities in inhibiting chromosome biorientation and in recruiting RZZ and dynein-dynactin came from our analysis of the BUB-1$^{ΔKD}$ mutant, which could support the normal recruitment of DNC-2$^{p50}$ (*Figure 4—figure supplement 1F*) but did not inhibit chromosome biorientation in absence of HCP-1/2$^{CENP-F}$ (*Figure 3B–D*). Therefore, kinetochore recruitment of RZZ and dynein-dynactin by BUB-1 is important for initial end-on attachments, but does not explain BUB-1 inhibition of chromosome biorientation.

However, in contrast to the full BUB-1 depletion, the BUB-1$^{ΔKD}$ mutant also led to a decreased frequency of merotelic attachments (*Figures 3D* and *4E*) and reduced kinetochore stretches in absence of HCP-1/2$^{CENP-F}$ (*Figure 4—figure supplement 1G*). This suggests that the high degree of merotely observed upon BUB-1 codepletion with HCP-1/2$^{CENP-F}$ is largely caused by the loss of dynein from kinetochores, and that BUB-1 favours amphitelic attachments by recruiting dynein to

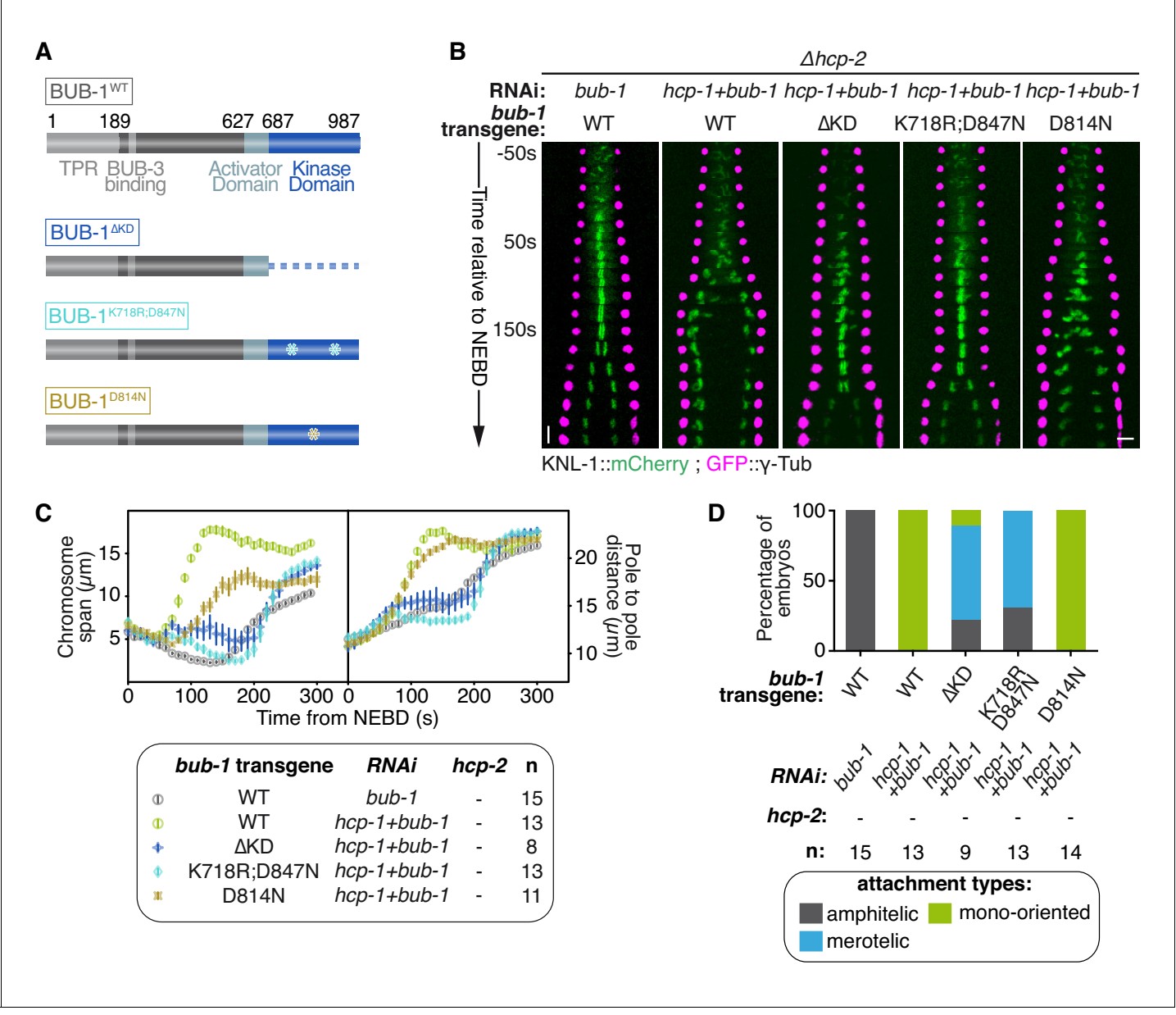

**Figure 3.** BUB-1 kinase domain inhibits biorientation independently of its kinase activity. (A) Schematics of wild-type (WT) BUB-1 and of the different BUB-1 mutants. (B) Kymographs generated from embryos expressing GFP::γ-Tub and KNL-1::mCherry, for different indicated conditions. (C) Chromosome span and pole to pole distance as functions of time after NEBD for the indicated conditions. (D) Quantification of the percentage of embryos with chromosomes engaged in amphitelic, merotelic and mono-oriented attachments in the indicated conditions. Error bars represent the SEM. Horizontal scale bar, 5 μm; Vertical scale bar, 20 s.
DOI: https://doi.org/10.7554/eLife.40690.018

The following source data and figure supplements are available for figure 3:

**Source data 1.** Chromosome span and pole to pole distance as functions of time after NEBD for the indicated conditions.
DOI: https://doi.org/10.7554/eLife.40690.023
**Source data 2.** Percentage of embryos with chromosomes engaged in amphitelic, merotelic and mono-oriented attachments, in the indicated conditions.
DOI: https://doi.org/10.7554/eLife.40690.024
**Figure supplement 1.** BUB-1 kinase domain inhibits biorientation independently of its kinase activity.
DOI: https://doi.org/10.7554/eLife.40690.019
**Figure supplement 1—source data 1.** Chromosome span and pole to pole distance as functions of time after NEBD for the indicated conditions.
DOI: https://doi.org/10.7554/eLife.40690.020
*Figure 3 continued on next page*

*Figure 3 continued*

**Figure supplement 1—source data 2.** BUB-1::mCherry and GFP::HCP-1CENP-F signals measured on kinetochores at metaphase in the indicated conditions.
DOI: https://doi.org/10.7554/eLife.40690.021

**Figure supplement 1—source data 3.** Percentage of embryos with chromosomes engaged in amphitelic and merotelic, in the indicated conditions.
DOI: https://doi.org/10.7554/eLife.40690.022

kinetochores. Furthermore, the spindle pole elongation profile observed in presence of BUB-1$^{\Delta KD}$ (and also in presence of BUB-1$^{K718R\ ;D847N}$ or BUB-1$^{D814N}$) did not display the typical bump indicative of delayed load-bearing attachments, normally observed in absence of BUB-1 (*Figure 3—figure supplement 1B*) (*Kim et al., 2015*). Altogether, these results suggest that recruiting the RZZ complex and dynein-dynactin at the kinetochore is a critical function of BUB-1 to promote amphitelic end-on attachments (*Gassmann et al., 2008a*). However, this activity is not sufficient to account for BUB-1 inhibitory effect on chromosome biorientation in absence of HCP-1/2$^{CENP-F}$.

## BUB-1 inhibits biorientation in absence of HCP-1/2$^{CENP-F}$ by preventing SKA complex recruitment

Another hypothesis to explain how BUB-1 inhibits chromosome biorientation could be through the limitation of attachment stability or strength, which would prevent chromosomes from stably engaging in load-bearing connections to both spindle poles. An obvious candidate that destabilises attachments is Aurora B (*Figure 5—figure supplement 1A*), which is recruited to centromeres downstream of BUB1 in human tissue cultured cells (*Ricke et al., 2012*). However, previous studies in human cells have ruled out a role for Aurora B in BUB1 function during chromosome alignment (*Logarinho et al., 2008*; *Meraldi and Sorger, 2005*; *Raaijmakers et al., 2018*). We confirmed these results in *C. elegans* embryos by using a fast-acting temperature-sensitive (ts) AIR-2$^{Aurora\ B}$ mutant (*air-2(or207ts), hereafter air-2(ts)*) and by upshifting embryos to the restrictive temperature (26°C) as pronuclei finished migrating (*Davies et al., 2017*; *Severson et al., 2000*). Upon upshift of HCP-1/2$^{CENP-F}$-depleted *air-2(ts)* mutant embryos to the restrictive temperature, sister chromatids still co-segregated prematurely (100 s after NEBD). Therefore BUB-1 inhibits chromosome congression and biorientation independently of AIR-2$^{Aurora\ B}$ activity (*Figure 5—figure supplement 1B,C*).

If BUB-1 does not promote attachment destabilization, it could instead counteract a stabilizing mechanism. In human cells, BUB1 localization to kinetochores is anti-correlated with SKA1 localization, a member of the SKA complex that strengthens attachment and subsequent congression of chromosomes (*Auckland et al., 2017*; *Schmidt et al., 2012*). BUB-1 might therefore inhibit biorientation by limiting the recruitment of the SKA complex to kinetochores (*Figure 5A*). Accordingly, we found that GFP-tagged SKA-1 was undetectable at kinetochores in absence of HCP-1/2$^{CENP-F}$ (*Figure 5B*; *Figure 5—figure supplement 2A,B*). However, as SKA-1 kinetochore localization depends on the tension exerted at kinetochores by end-on attachments (*Cheerambathur et al., 2017*), the low tension generated at mono-oriented chromosomes could explain the lack of SKA-1 kinetochore localization in this condition. To exclude

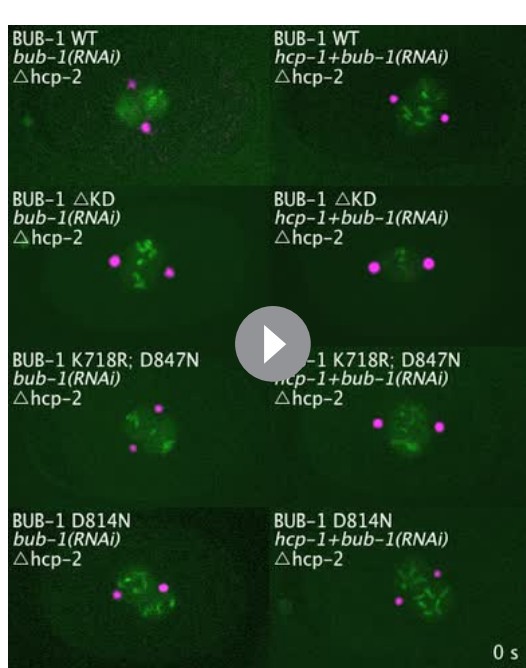

**Video 3.** One-cell *C. elegans* embryos in the indicated conditions. 10 s per frame. Magenta, γ-Tubulin::GFP (spindle poles); Green, KNL-1::mCherry (kinetochores).
DOI: https://doi.org/10.7554/eLife.40690.025

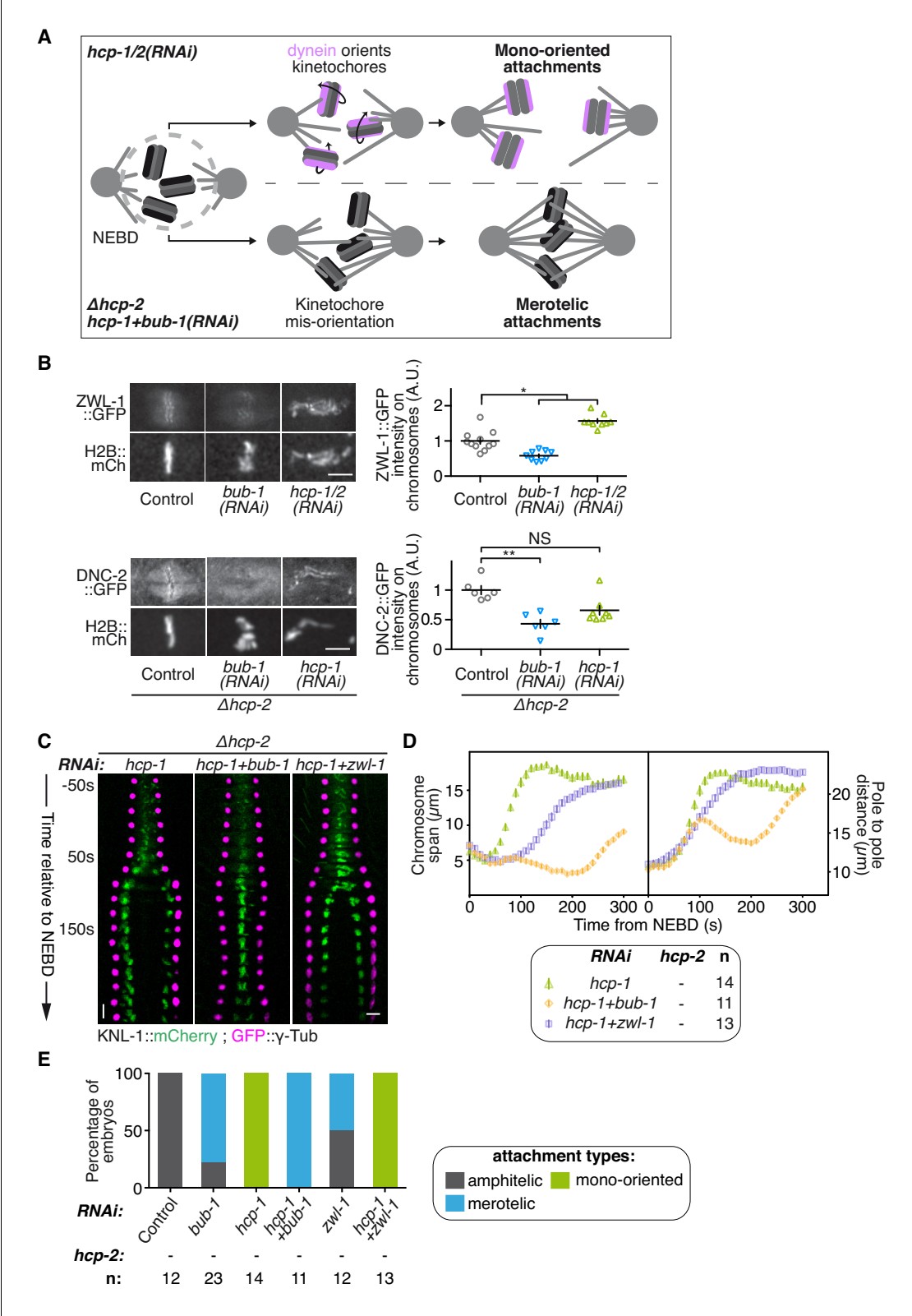

**Figure 4.** BUB-1 accelerates non-merotelic end-on attachments via dynein, independently of the inhibitory effect on biorientation. (**A**) Schematics of the potential mechanism for chromosome biorientation inhibition by BUB-1 in absence of HCP-1/2$^{CENP-F}$. By orienting kinetochores relative to spindle poles, BUB-1 downstream partners RZZ and dynein-dynactin connect chromosomes in a mono-oriented conformation to short non-dynamic microtubules, leading to premature chromosome segregation. In absence of BUB-1, kinetochore mis-orientation enables the establishment of

*Figure 4 continued on next page*

*Figure 4 continued*

biorented merotelic connections capable of resisting cortical traction forces. (**B**) Left: Representative images from time-lapse movies showing localisations of ZWL-1$^{Zwilch}$::GFP and DNC-2::GFP on chromosomes (H2B::mCherry) in the indicated conditions, 100 s after NEBD. Right: Quantifications of the GFP signal on chromosomes 100 s after NEBD. Kruskall Wallis tests with Dunn's correction for multiplicity were used to assess significance (ZWL-1::GFP bub-1(RNAi) p = 0,0276, ZWL-1::GFP hcp-1/2(RNAi) p = 0,0341, DNC-2::GFP bub-1(RNAi) p = 0,0015, DNC-2::GFP hcp-1(RNAi) p = 0,0671). (**C**) Kymographs generated from embryos expressing GFP::γ-Tub and KNL-1::mCherry for the indicated conditions. (**D**) Chromosome span and pole to pole distance as functions of time after NEBD for the indicated conditions. (**E**) Quantification of the percentage of embryos with chromosomes engaged in amphitelic, merotelic and mono-oriented attachments in the indicated conditions. Error bars represent the SEM. Horizontal scale bars, 5 μm; Vertical scale bar, 20 s.

DOI: https://doi.org/10.7554/eLife.40690.026

The following source data and figure supplements are available for figure 4:

**Source data 1.** ZWL-1Zwilch::GFP and DNC-2::GFP signals on chromosomes 100 safter NEBD.
DOI: https://doi.org/10.7554/eLife.40690.031
**Source data 2.** Chromosome span and pole to pole distance as functions of time after NEBD for the indicated conditions.
DOI: https://doi.org/10.7554/eLife.40690.032
**Source data 3.** Percentage of embryos with chromosomes engaged in amphitelic, merotelic and mono-oriented attachments, in the indicated conditions.
DOI: https://doi.org/10.7554/eLife.40690.033
**Figure supplement 1.** BUB-1 accelerates non-merotelic end-on attachments via dynein, independently of the inhibitory effect on biorientation.
DOI: https://doi.org/10.7554/eLife.40690.027
**Figure supplement 1—source data 1.** Integrated ZWL-1::GFP signal on chromosomes as a function of time for the indicated conditions.
DOI: https://doi.org/10.7554/eLife.40690.028
**Figure supplement 1—source data 2.** Integrated DNC-2::GFP signal on chromosomes as a function of time for the indicated conditions.
DOI: https://doi.org/10.7554/eLife.40690.029
**Figure supplement 1—source data 3.** DNC-2::GFP signal on kinetochores 100 safter NEBD in the indicated conditions.
DOI: https://doi.org/10.7554/eLife.40690.030

this possibility, we generated monopolar spindles in two-cell stage embryos by depleting ZYG-1$^{Plk4}$, which is essential for centrosome duplication and subsequent bipolar spindle assembly (*Essex et al., 2009*; *O'Connell et al., 2001*). In this condition, SKA-1::GFP accumulated on the chromosome side that faces the spindle monopole, demonstrating that mono-oriented attachments in ZYG-1$^{Plk4}$-depleted 2 cell stage embryos provide sufficient tension for SKA complex kinetochore localization (*Figure 5B*; *Figure 5—figure supplement 2A,C*). Together, these results suggest that SKA-1 kinetochore recruitment is inhibited in absence of HCP-1/2$^{CENP-F}$, but that the mono-oriented attachment configuration is not sufficient to account for the lack of kinetochore SKA-1. Most importantly, SKA-1::GFP kinetochore recruitment was rescued by depleting BUB-1 in absence of HCP-1/2$^{CENP-F}$ (*Figure 5B*; *Figure 5—figure supplement 2A,B*), suggesting that BUB-1 prevents SKA-1 recruitment to kinetochores when HCP-1/2$^{CENP-F}$ is absent.

We next wanted to test if this novel activity of BUB-1 could account for its inhibitory effect on chromosome biorientation. If BUB-1 inhibition of SKA complex kinetochore recruitment prevents chromosome biorientation in absence of HCP-1/2$^{CENP-F}$, two predictions can be made: 1) BUB-1 mutants incapable of inhibiting chromosome biorientation should not prevent SKA complex kinetochore recruitment, and 2) the absence of the SKA complex should abrogate chromosome biorientation in these mutants. To test these predictions, we focused on the BUB-1$^{ΔKD}$ and BUB-1$^{K718R\ ;D847N}$ mutants, which are incapable of inhibiting chromosome biorientation, while promoting proper initial end-on attachment through kinetochore recruitment of RZZ and dynein-dynactin. In agreement with our first prediction, both mutants allowed kinetochore recruitment of SKA-1::GFP (*Figure 5—figure supplement*

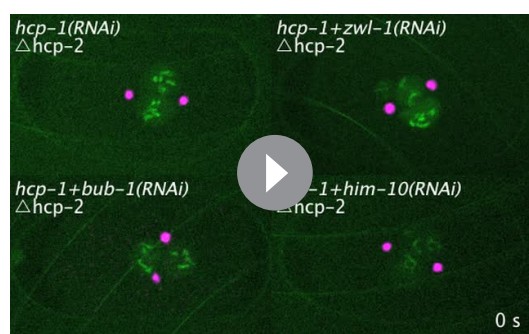

**Video 4.** One-cell *C. elegans* embryos in the indicated conditions. 10 s per frame. Magenta, γ-Tubulin::GFP (spindle poles); Green, KNL-1::mCherry (kinetochores).
DOI: https://doi.org/10.7554/eLife.40690.034

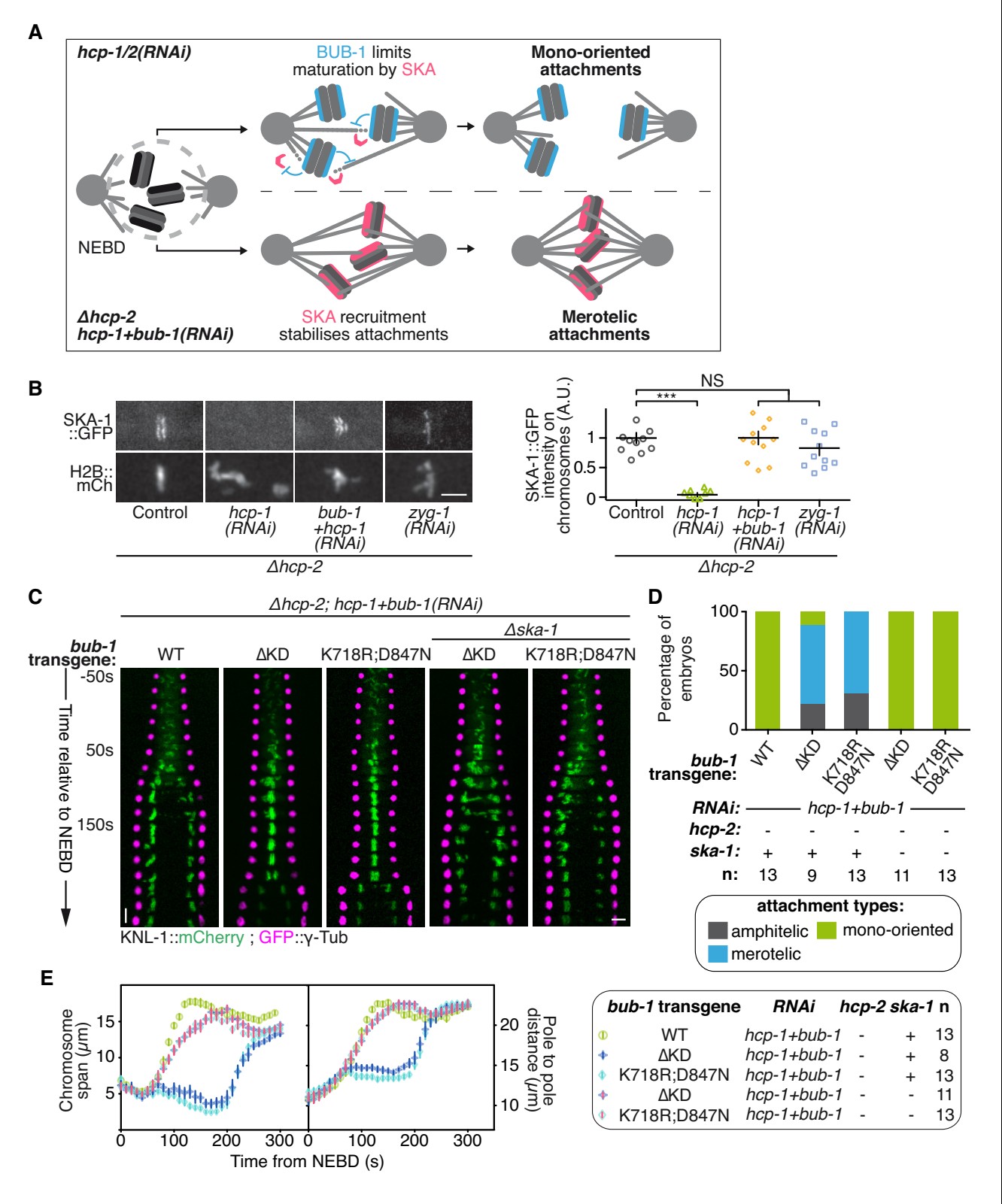

**Figure 5.** BUB-1 inhibits biorientation in absence of HCP-1/2$^{CENP-F}$ by preventing SKA complex recruitment. (**A**) Schematics of the potential mechanism for chromosome biorientation inhibition by BUB-1 in absence of HCP-1/2$^{CENP-F}$. BUB-1 limits attachment maturation by the SKA complex, leading to the incapacity for chromosomes to biorient when microtubules are short and non-dynamic. Co-depleting BUB-1 restores SKA complex recruitment, allowing the strengthening of attachments and therefore the establishment of biorientation even when microtubules are short and non-dynamic. (**B**)
*Figure 5 continued on next page*

*Figure 5 continued*

Left: Representative images from time-lapse movies showing the localization of SKA-1::GFP on chromosomes (H2B::mCherry) in the indicated conditions at metaphase. Right: Quantification of the GFP signal on chromosomes at metaphase. Kruskall Wallis tests with Dunn's correction for multiplicity were used to assess significance (hcp-1(RNAi) p = 0,0006, hcp-1 +bub-1(RNAi) p > 0.9999, zyg-1(RNAi) p > 0,9999). (C) Kymographs generated from embryos expressing GFP::γ-Tub and KNL-1::mCherry for the indicated conditions. (D) Quantification of the percentage of embryos with chromosomes engaged in amphitelic, merotelic and mono-oriented attachments in the indicated conditions. (E) Chromosome span and pole to pole distance as functions of time after NEBD for the indicated conditions. Error bars represent the SEM. Horizontal scale bars, 5 μm; Vertical scale bar, 20 s.
DOI: https://doi.org/10.7554/eLife.40690.035

The following source data and figure supplements are available for figure 5:

**Source data 1.** SKA-1::GFP signal on chromosomes at metaphase.
DOI: https://doi.org/10.7554/eLife.40690.044

**Source data 2.** Percentage of embryos with chromosomes engaged in amphitelic, merotelic and mono-oriented attachments, in the indicated conditions.
DOI: https://doi.org/10.7554/eLife.40690.045

**Source data 3.** Chromosome span and pole to pole distance as functions of time after NEBD for the indicated conditions.
DOI: https://doi.org/10.7554/eLife.40690.046

**Figure supplement 1.** BUB-1 does not inhibit biorientation via AIR-2$^{AuroraB}$.
DOI: https://doi.org/10.7554/eLife.40690.036

**Figure supplement 1—source data 1.** Chromosome span and pole to pole distance as functions of time after NEBD for the indicated conditions.
DOI: https://doi.org/10.7554/eLife.40690.037

**Figure supplement 2.** BUB-1 inhibits chromosome biorientation in absence of HCP-1/2$^{CENP-F}$ by preventing SKA complex recruitment.
DOI: https://doi.org/10.7554/eLife.40690.038

**Figure supplement 2—source data 1.** SKA-1::GFP signal measured on chromosomes as a function of time.
DOI: https://doi.org/10.7554/eLife.40690.039

**Figure supplement 2—source data 2.** SKA-1::GFP and H2B::mCherry intensities along a linescan.
DOI: https://doi.org/10.7554/eLife.40690.040

**Figure supplement 2—source data 3.** Chromosome span and pole to pole distance as functions of time after NEBD for the indicated conditions.
DOI: https://doi.org/10.7554/eLife.40690.041

**Figure supplement 2—source data 4.** SKA-1::GFP signal on chromosomes at metaphase in the indicated conditions.
DOI: https://doi.org/10.7554/eLife.40690.042

**Figure supplement 2—source data 5.** Integrated BUB-1::GFP signal measured on chromosomes as a function of time.
DOI: https://doi.org/10.7554/eLife.40690.043

*2F*). Next, to test if this kinetochore localization of the SKA complex is required for chromosome biorientation in the absence of HCP-1/2$^{CENP-F}$ in these BUB-1 mutants, we used a *ska-1* deletion allele (Δ*ska-1*). Although required for timely chromosome congression, SKA-1 is a non-essential gene in *C. elegans* (*Figure 5—figure supplement 2D,E*) (*Cheerambathur et al., 2017*). Strikingly in the simultaneous absence of endogenous BUB-1, HCP-1/2$^{CENP-F}$ and SKA-1, both BUB-1 mutants (BUB-1$^{ΔKD}$ and BUB-1$^{K718R ;D847N}$) led to complete failure of chromosome congression and biorientation, along with sister chromatid co-segregation, similar to what is observed after HCP-1/2$^{CENP-F}$ depletion alone (*Figure 5C–E*; *Figure 5—figure supplement 2G*; *Video 5*). Thus BUB-1$^{ΔKD}$ and BUB-1$^{K718R ;D847N}$ are capable of sustaining chromosome biorientation in absence of HCP-1/2$^{CENP-F}$ only when the SKA complex is present. This result suggests that in these two BUB-1 mutants the kinetochore-localized SKA complex is responsible for the observed rescue of chromosome biorientation in absence of HCP-1/2$^{CENP-F}$. Altogether, our results suggest that, in absence of HCP-1/2$^{CENP-F}$ or CLS-2$^{CLASP}$, BUB-1 inhibits stable biorientation of chromosomes by preventing SKA complex recruitment to kinetochores (*Figure 5A*).

## BUB-1 limits merotely by balancing kinetochore microtubule assembly and kinetochore attachment maturation

We next wanted to test the role of BUB-1 downstream activities for accurate chromosome

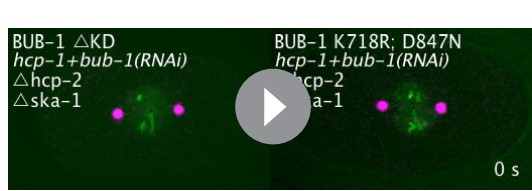

**Video 5.** One-cell *C. elegans* embryos in the indicated conditions. 10 s per frame. Magenta, γ-Tubulin::GFP (spindle poles); Green, KNL-1::mCherry (kinetochores).
DOI: https://doi.org/10.7554/eLife.40690.047

segregation and organismal viability. Upon loss of CLS-2$^{CLASP}$ -a protein involved in promoting kinetochore microtubule polymerization- *C. elegans* zygotes present a unique phenotype of premature sister chromatid co-segregation (*Cheeseman et al., 2005*). We found that this phenotype can be rescued by alleviating BUB-1-mediated inhibition of SKA-1 at the kinetochores, preventing SKA-mediated attachment maturation. These observations suggest that the capacity for chromosomes to engage in stable connections to both spindle poles can be provided either by efficient kinetochore microtubule polymerization or by kinetochore-microtubule attachment maturation. Thus, BUB-1 provides two opposing activities in the establishment of bioriented amphitelic connections: (1) promoting biorientation by recruiting CLS-2$^{CLASP}$ to kinetochores, and (2) restricting biorientation by limiting SKA-mediated attachment maturation. To probe the functional significance of these opposing BUB-1 activities for accurate chromosome segregation and overall organismal viability, we used the BUB-1 mutants that separate these opposing activities. We used the BUB-1$^{\Delta KD}$ mutant that does not recruit HCP-1/2$^{CENP-F}$ and CLS-2$^{CLASP}$ to kinetochores, but nevertheless allows chromosome biorientation due to its incapacity to limit SKA-mediated attachment maturation, to probe the importance of BUB-1 activity in promoting kinetochore microtubule assembly through CLS-2$^{CLASP}$ kinetochore recruitment. Embryos expressing BUB-1$^{\Delta KD}$ only presented slight merotely at 24℃, but lowering the temperature down to 15℃, and therefore lowering overall microtubule growth (*Srayko et al., 2005*), raised this rate of merotely by three-fold (*Figure 6A*). We observed a similar temperature dependency for embryonic lethality in this mutant (*Figure 6B*). These results suggest that promoting kinetochore microtubule assembly via BUB-1-mediated recruitment of CLS-2$^{CLASP}$ is required to avoid merotely and embryonic lethality. This is consistent with previous work in human cancer cells with chromosomal instability (*Bakhoum et al., 2009a*).

We finally addressed the importance of limiting SKA-dependent attachment maturation in the context of dynamic kinetochore microtubules (when HCP-1/2$^{CENP-F}$ and CLS-2$^{CLASP}$ are present at the kinetochore). The BUB-1$^{K718R\ ;D847N}$ mutant does not limit SKA-mediated attachment maturation but still recruits HCP-1/2$^{CENP-F}$ and CLS-2$^{CLASP}$ to kinetochores. In contrast to control BUB-1$^{WT}$-expressing embryos, we found that BUB-1$^{K718R\ ;D847N}$ mutant-expressing embryos displayed up to 21% embryonic lethality when raised at 24℃. Embryos depleted of the SAC protein MDF-1$^{Mad1}$ did not show the same temperature-dependent lethality (8%), nor did the BUB-1$^{D814N}$ mutant (3.4%). The BUB-1$^{K718R\ ;D847N}$ mutant phenotype is therefore not caused by a defective SAC or a lack of BUB-1 kinase activity, but rather by the enhanced microtubule growth at high temperature (*Figure 6C*). In other words, when load-bearing connections are favoured by high microtubule dynamics, limiting SKA-mediated attachment maturation becomes essential. Together, our data suggest that BUB-1 displays opposing activities at the kinetochore to promote properly bioriented kinetochore-microtubule attachments and to prevent embryonic lethality.

## Discussion

The conserved kinase BUB1 is a SAC protein that is also involved in kinetochore-microtubule attachments and proper chromosome segregation in diverse species. However, the molecular mechanisms of this non-SAC role for BUB1 remain elusive. Here, we identify two previously uncharacterized functions of the *C. elegans* BUB-1 kinase in (1) accelerating end-on kinetochore-microtubule attachments by recruiting the dynein-dynactin complex to kinetochores, and (2) limiting attachment maturation by the SKA complex (*Figure 6E*). We show this latter activity relies on the BUB-1 kinase domain but is independent of kinase activity. Previously, *C. elegans* BUB-1 was shown to target the two redundant CENP-F orthologs HCP-1/2$^{CENP-F}$ to kinetochores, which in turn recruit the CLASP protein CLS-2$^{CLASP}$ (*Cheeseman et al., 2005*; *Encalada et al., 2005*). CLASP at kinetochores promotes microtubule assembly and increases microtubule dynamics, favouring amphitelic attachments (*Maffini et al., 2009*; *Maiato et al., 2003a*; *Maiato et al., 2005*). Therefore, our results show that BUB-1 both promotes amphitely through the RZZ-dynein-dynactin complex and CLS-2$^{CLASP}$, while limiting maturation of chromosome biorientation by the SKA complex. Our data also suggest that the coordination of these different activities by BUB-1 is required to avoid merotely and embryonic lethality. These results therefore define a central role for BUB-1 in promoting accurate chromosome biorientation essential to maintain genetic integrity during cell division.

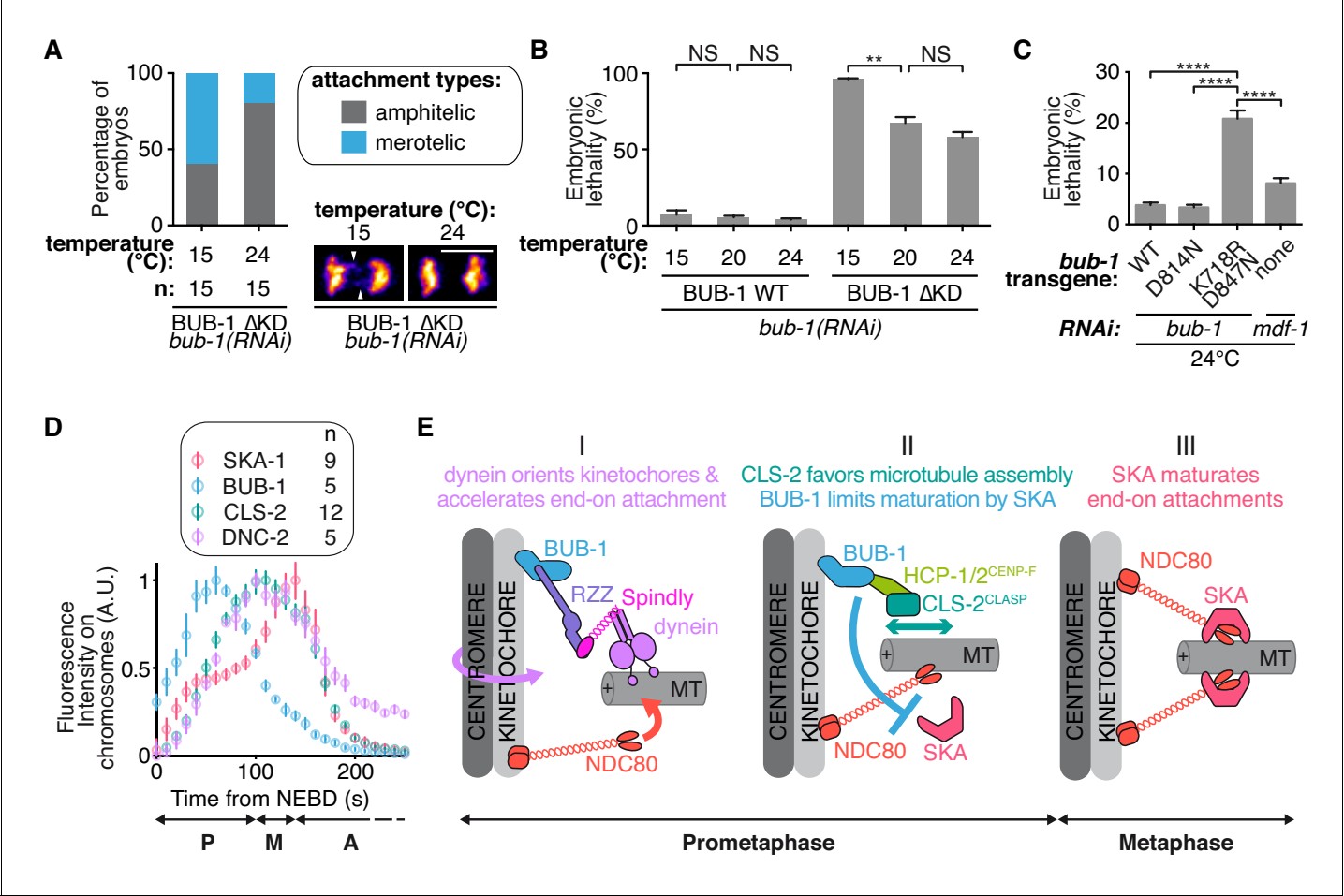

**Figure 6.** BUB-1 limits merotely by balancing kinetochore microtubule assembly and kinetochore attachment maturation. (**A**) Quantification of the percentage of embryos with chromosomes engaged in amphitelic and merotelic attachments in the indicated conditions. Representative images of merotelic and non merotelic segregations are shown based on the KNL-1::mCherry signal 20 s after anaphase onset. White arrowheads point toward kinetochore stretches. (**B**) Embryonic lethality in the indicated conditions. Kruskall Wallis tests with Dunn's correction for multiplicity were used to assess significance (WT 15°C (n = 252 embryos) vs 20°C (n = 1934) p > 0,9999, WT 20°C vs 24°C (n = 2107) p > 0,9999, ΔKD 15°C (n = 352) vs 20°C (n = 1104) p = 0,0012, ΔKD 20°C vs 24°C (n = 751) p = 0,4722). (**C**) Embryonic lethality in the indicated conditions. Kruskall Wallis tests with Dunn's correction for multiplicity were used to assess significance (WT (n = 2107 embryos) vs K718R;D847N (n = 2663) p < 0,0001, D814N (n = 2181) vs K718R;D847N p < 0,0001, K718R;D847N vs *mdf-1(RNAi)* (n = 3410) p < 0,0001). (**D**) Quantifications of the integrated signals measured on chromosomes for different GFP-tagged proteins as function of time from NEBD. (**E**) (I) In prometaphase, BUB-1 localises to kinetochores and favours amphitely via two independent mechanisms. Downstream of BUB-1, the RZZ complex, Spindly and dynein-dynactin orient kinetochores and regulate NDC-80 activity, leading to the acceleration of end-on attachment in a non-merotelic conformation. (II) BUB-1 further contributes to establishing amphitelic attachments by promoting kinetochore microtubule assembly via HCP-1/2[CENP-F] and CLS-2[CLASP] recruitment, while limiting attachment maturation via the SKA complex. (III) In metaphase, BUB-1 leaves kinetochores allowing attachment maturation by the SKA complex. Scale bar, 5 μm.

DOI: https://doi.org/10.7554/eLife.40690.048

The following source data is available for figure 6:

**Source data 1.** Percentage of embryos with chromosomes engaged in amphitelic and merotelic attachments in the indicated conditions.
DOI: https://doi.org/10.7554/eLife.40690.049
**Source data 2.** Percentage of embryonic lethality in the indicated conditions.
DOI: https://doi.org/10.7554/eLife.40690.050
**Source data 3.** Percentage of embryonic lethality in the indicated conditions.
DOI: https://doi.org/10.7554/eLife.40690.051
**Source data 4.** Integrated signals measured on chromosomes for the indicated GFP-tagged proteins as function of time.
DOI: https://doi.org/10.7554/eLife.40690.052

## BUB-1-mediated kinetochore recruitment of RZZ and dynein-dynactin

BUB1 contributes to RZZ recruitment in human cells (*Caldas et al., 2015*; *Zhang et al., 2015*), how-ever previously this function has not been linked to the role of BUB1 in chromosome alignment and segregation. Here, we show that in *C. elegans*, BUB-1 also contributes to kinetochore recruitment of the RZZ complex and that this is essential to avoid merotely. In human cells, some RZZ remains at kinetochores following BUB1 depletion, which suggests an additional kinetochore-binding site must exist. The kinetochore protein Zwint, which interacts with the KNL1 C-terminal half, was originally proposed to be the primary kinetochore-docking partner of RZZ in human cells (*Kops et al., 2005*; *Petrovic et al., 2010*; *Starr et al., 2000*; *Wang et al., 2004*). However, some RZZ remains at kineto-chores in human cells even in absence of KNL1 (and thus also in absence of kinetochore-localized Zwint) and BUB1. Consistently, the distant Zwint ortholog KBP-5 is dispensable for RZZ kinetochore recruitment in *C. elegans* (*Varma et al., 2013*). A Zwint-, KNL1- and BUB1-independent pathway has therefore been hypothesized to participate in RZZ localization at human kinetochores (*Caldas et al., 2015*; *Silió et al., 2015*; *Zhang et al., 2015*). Although the molecular details of this alternative con-tact site between RZZ and the kinetochore are still unclear, our results are consistent with this hypothesis, as BUB-1 depletion does not fully abrogate kinetochore RZZ localization (*Figure 4B*, *Fig-ure 4—figure supplement 1A,B*).

In contrast, we found that BUB-1 is absolutely required for dynein-dynactin recruitment to kineto-chores. One potential explanation for the apparent discrepancy between this complete lack of dynein-dynactin at kinetochores despite partial recruitment of the RZZ complex may stem from BUB-1 downstream partners HCP-1/2$^{CENP-F}$, which recruit NUD-2$^{NudE/el}$ to stabilise dynein-dynactin at kinetochores (*Simões et al., 2018*). BUB-1 would therefore both recruit dynein-dynactin via the RZZ complex, and stabilise it at kinetochores via HCP-1/2$^{CENP-F}$ and NUD-2$^{NudE/el}$.

Our functional analysis comparing BUB-1-depleted embryos in the presence or absence of the BUB-1$^{\Delta KD}$ mutant, which recruits DNC-2$^{p50}$ normally to kinetochores (*Figure 4—figure supplement 1F*), shows that both BUB-1-mediated RZZ and dynein-dynactin recruitments contribute to accelerat-ing the establishment of end-on attachments and to the prevention of merotely (*Figure 4—figure supplement 1G*). RZZ and dynein-dynactin therefore constitute a kinetochore module, recruited downstream of BUB-1, that contributes to rapidly and accurately establishing biorientation in early mitosis.

## BUB-1 control of microtubule dynamics and attachment maturation

In *C. elegans* zygotes, BUB-1 targets HCP-1/2$^{CENP-F}$ to kinetochores, which in turn recruits CLS-2$^{CLASP}$ (*Cheeseman et al., 2005*; *Encalada et al., 2005*). Kinetochore-localized CLS-2$^{CLASP}$ is thought to promote microtubule polymerization essential for proper chromosome biorientation. Our work stresses the importance of BUB-1 in antagonizing CLS-2$^{CLASP}$-dependent kinetochore microtu-bule assembly by limiting their attachment to kinetochores. We indeed found that upon HCP-1/2$^{CENP-F}$ or CLS-2$^{CLASP}$ depletion, chromosomes become bioriented only when kinetochore-localized BUB-1 is absent. Our results further suggest that BUB-1 limits kinetochore-microtubule attachment maturation by preventing SKA complex kinetochore recruitment.

How BUB-1 limits the maturation of kinetochore microtubule attachments by the SKA complex remains unclear. We envision several non-exclusive possibilities. First, BUB-1 could inhibit SKA com-plex kinetochore recruitment through a direct physical interaction, although we currently do not have evidence for this premise. Second, BUB-1 could indirectly regulate the interaction between the SKA complex and its kinetochore docking partner NDC-80, by recruiting Aurora A at the kineto-chore. Indeed, although Aurora B-dependent phosphorylation of the NDC-80 tail prevents enrich-ment of the SKA complex at kinetochores in *C. elegans* (*Cheerambathur et al., 2017*), we rule out here a role for Aurora B as a mediator of BUB-1 inhibition of biorientation (*Figure 5—figure supple-ment 1*). In contrast, in human cells, BUB1 was recently shown to recruit Aurora A at the centromere (*Eot-Houllier et al., 2018*), where it phosphorylates the NDC80 tail (*DeLuca et al., 2018*). Whether BUB-1-dependent recruitment of Aurora A also restricts SKA complex kinetochore localization remains to be investigated. Finally, BUB-1 could directly or indirectly decrease tension at kineto-chores. In *C. elegans*, kinetochore tension has been proposed to promote SKA complex recruitment (*Cheerambathur et al., 2017*). We showed that monopolar spindles, in which kinetochore tension is likely low, are compatible with normal level of SKA complex at kinetochores. However, mono-

oriented chromosomes, as visible in absence of HCP-1/2$^{CENP-F}$ or CLS-2$^{CLASP}$, could potentially display even lower kinetochore tension, which would abrogate SKA complex kinetochore localization. By increasing tension at kinetochores through an unknown mechanism, BUB-1 could restore SKA complex kinetochore enrichment and chromosome biorientation.

Regardless of the exact mechanism by which BUB-1 regulates the SKA complex, we surprisingly did not observe hyper-recruitment of SKA-1 to kinetochores in the absence of BUB-1 (*Figure 5—figure supplement 2A,B*), nor in the presence of BUB-1 mutants that do not inhibit chromosome biorientation in absence of HCP-1/2$^{CENP-F}$ (*Figure 5—figure supplement 2F*). However, upon depletion of HCP-1/2$^{CENP-F}$ (proteins dependent upon BUB-1 for kinetochore localisation, *Figure 1E*), when SKA complex kinetochore recruitment is inhibited, BUB-1 is hyper-recruited to kinetochores (*Figure 5—figure supplement 2H,I*). Although the reason for this hyper-recruitment is unclear, it suggests that a threshold level of BUB-1 could be required to prevent SKA complex localization at kinetochores. In this context, depleting HCP-1/2$^{CENP-F}$ or CLS-2$^{CLASP}$ would therefore amplify the BUB-1-mediated inhibition of SKA complex kinetochore localization, which in the *C. elegans* embryo is otherwise not apparent and probably extremely transient during prometaphase when biorientation is established. Nevertheless, the embryonic lethality observed in the BUB-1$^{K718R, D847N}$ mutant incapable of restricting biorientation (but which recruits HCP-1/2$^{CENP-F}$ and CLS-2$^{CLASP}$ at kinetochores) at high temperature (21% at 24°C) demonstrates the essential and physiological role of this pathway (*Figure 6A–C*).

## Temporal coordination of antagonistic activities at the kinetochore

Our study provides mechanistic insights into the non-SAC role for BUB-1 in chromosome segregation, suggesting it is a key regulator of the kinetochore-microtubule interface that facilitates amphitelic kinetochore-microtubule attachments and shields chromosomes from merotely. Based on these multiple BUB-1 functions and on the dynamic kinetochore localization of downstream components, we propose a temporal and functional framework for chromosome biorientation (*Figure 6D,E*). In this model, BUB-1 dependent recruitment of the RZZ complex and of dynein-dynactin favours initial end-on kinetochore-microtubule attachments in bioriented conformations. BUB-1 then controls the stability of these attachments through two opposing activities. First, BUB-1 favours NDC-80-mediated load-bearing attachments via recruitment of HCP-1/2$^{CENP-F}$ and CLS-2$^{CLASP}$, which promote kinetochore microtubule assembly. Second BUB-1 limits attachment maturation by preventing SKA complex kinetochore targeting. Together these activities help establishing amphitelic attachments and preventing errors in chromosome segregation. BUB-1 then progressively leaves from kinetochores in metaphase (*Figure 6D*; *Figure 5—figure supplement 2H,I*), allowing SKA complex accumulation and kinetochore microtubule attachment maturation. The high frequency of embryonic lethality observed in *C. elegans* zygotes in the presence of BUB-1 mutants that uncouple these two functions (BUB-1$^{\Delta KD}$ or BUB-1$^{K718R\ ;D847N}$), stresses the importance of coordinating temporally kinetochore microtubule initial attachments and their maturation.

In conclusion, using the *C. elegans* model, we reveal a non-SAC role for BUB-1 in limiting SKA-mediated attachment maturation, when HCP-1/2$^{CENP-F}$ or CLS-2$^{CLASP}$ is deficient. We suspect that this is a conserved and general role of BUB1. Indeed, we found that BUB-1-mediated inhibition of chromosome biorientation is linked to its kinase domain, but not to the kinase activity. This is also the case for the yet unidentified BUB1-dependent activity that prevents lethality of SAC-deficient haploid HAP1 human cells (*Raaijmakers et al., 2018*). In future work, it will be important to determine whether there is a functional link between BUB-1-mediated regulation of the SKA complex - that we identified here- and the role for BUB-1 in preventing lethality of SAC-deficient haploid human cells. The conservation of all the key molecular players suggests this non-SAC role for BUB-1 in regulating kinetochore-microtubule attachments could be a general and conserved function of BUB1 in metazoans.

## Materials and methods

### *C. elegans* strain maintenance

*Supplementary file 1A* lists the strains used throughout this study. Strains were maintained on nematode growth medium plates seeded with OP50 bacteria. They were incubated at 23°C, with the

exception of the *air-2(or207ts)* mutant that was maintained at the permissive temperature of 16°C. Most strains were generated by crossing previously existing *C. elegans* lines. The transgenes generated for the purpose of this study were engineered using CRISPR-Cas9 for the Δ*hcp-1* and Δ*hcp-2* strains (see below) and the GFP-tagged endogenous HCP-1 and HCP-2 strains (purchased from SunyBiotech), and using MosSCI for the BUB-1$^{ΔKD}$ mutant (*Katic et al., 2015*).

## RNA-mediated interference

*Supplementary file 1B* lists the dsRNAs used in this study, which were synthesized using the indicated primers and templates. After DNA amplification by PCR, reactions were cleaned (PCR purification kit, Qiagen), and used as templates for T3 and T7 transcription reactions (MEGAscript, Invitrogen) for 5 hr at 37°C. These reactions were cleaned (MEGAclear, Invitrogen), then combined for annealing at 68°C for 10 min and 37°C for 30 min. L4 larvae were injected at the indicated concentrations in the pseudo-coelum, and incubated for 48 hr at 20°C, or 16°C for the *air-2(or207ts)* mutant.

## CRISPR-Cas9 mutant generation

The mutations in the hcp-1 and hcp-2 loci were generated by using a CRISPR-Cas9 approach previously described (*Katic et al., 2015*). This strategy aims at generating a deletion between two double-strand breaks surrounding the start codon, and at inserting a screening cassette by Non-Homologous End-Joining. sgRNAs were designed using the http://crispr.mit.edu/ design tool, and cloned in the pIK198 vector under the control of the *C. elegans* U6 promoter (Injected at 50 ng/μL). *Supplementary file 1C* lists the sgRNA targets, and their positions relative to the *hcp-1* and *hcp-2* loci. The cas-9 protease was provided as a cDNA under the eft-3 promoter in the pDD122 vector (50 ng/μL). To help screening for mutations, *unc-119(ed3)* mutant worms were injected, and the injection mix provided a vector containing an *unc-119* rescue cassette (50 ng/μL). In order to linearize this vector, the sequence and PAM targeted by the sgRNA downstream of the ATG were inserted in the vector. In addition, fluorescence markers encoded in vectors pCFJ90 (Pmyo-2::mCherry, 2.5 ng/μL), pCFJ104 (Pmyo-3::mCherry, 5 ng/μL) and pGH8 (Prab-3::mCherry, 5 ng/μL), were also injected. Worms were injected on day 0, cloned, and left to starve. Plates with moving worms were selected and chunked at day 14. Worms expressing the fluorescent co-injection markers were negatively selected on day 15 as worms containing extra-chromosomal arrays. Remaining worms were then cloned, and their progeny was screened by PCR.

## Western blotting

HCP-1 and HCP-2 expression in the Δ*hcp-1 and* Δ*hcp-2* strains was assessed by immunoblotting. For each well, 50 worms were washed in M9 (22 mM KH$_2$PO$_4$, 42 mM Na$_2$HPO$_4$, 86 mM NaCl, and 1 mM MgSO$_4$•7H$_2$O) supplemented with 0.1% Triton X-100. Worms were then resuspended in 20 μL sample buffer (40% glycerol, 240 mM Tris-HCl, pH 6.8, 8% SDS, 0.04% bromophenol blue, and 5% β-mercaptoethanol) before being boiled for 15 min at 95°C, vortexed for 15 min, and boiled again for 15 min at 95°C. Extracts were then loaded on a NuPAGE 3–8% TrisAcetate Gel (Invitrogen), transferred to nitro-cellulose, and incubated with 1 μg/mL antibodies targeting either HCP-1 C-Terminal domain, HCP-2 N-Terminal or C-Terminal domains. A mouse anti-α-tubulin antibody (DM1α, Abcam) was used as a loading control.

## Live imaging and image analysis

Worms were dissected to free embryos, which were then mounted and imaged as described in (*Laband et al., 2018*). Acquisitions were made on a spinning disk confocal microscope (Roper Scientific), using a CFI APO LBDA S × 60/NA1.4 oil objective and a CoolSNAP HQ2 CCD camera (Photometrics Scientific). All movies were acquired with a 2 × 2 binning. Metamorph seven software (Molecular devices) was used for control of acquisition parameters. Unless specified otherwise, all movies were acquired at temperatures varying between 22°C and 24°C. The CherryTemp temperature controller system (CherryBiotech) was used to perform imaging experiments at the specific temperatures of 15°C, 24°C and 26°C. Image analysis was performed using Fiji (*Schindelin et al., 2012*) and the Python scikit-image library (*van der Walt et al., 2014*).

## Pole to pole distance and chromosome span measurements

Embryos expressing γ-Tubulin::GFP and KNL-1::mCherry were imaged at 10 s intervals, with 3 z slices spaced 2 µm apart for the fluorescence channels, and a single slice per time point for the DIC frame. Measurements were carried out on maximal z-projections of the fluorescence channels. Pole to pole distance was measured using a Python script to segment poles in the GFP channel, and measure distances between their centroids. Chromosome span was measured manually by tracing the shortest line in the spindle pole axis comprising the totality of the KNL-1::mCherry kinetochore signal. Measurements were then aligned with reference to NEBD detected in the DIC channel.

## Fluorescence intensity measurements

Fluorescently tagged protein quantifications (GFP::HCP-1, GFP::HCP-2, CLS-2::GFP, HCP-3HFD:: CPAR-1Ntail::GFP, ZWL-1::GFP, DNC-2::GFP, SKA-1::GFP, and BUB-1::GFP) were generated by imaging embryos at 10 s intervals, with 4 z slices spaced 2 µm apart for the fluorescence channels, and a single slice per time point in DIC. This z sampling doesn't allow the measurement of the integrated signal throughout the spindle, but ensures the spindle, which moves within the embryo, is always in focus in at least one of the slices. Measurements were therefore carried out on maximal z-projections of the GFP channel. A Python script was used to segment chromosomes in the mCherry channel, and allowed access to the integrated GFP signal on chromosomes over time. For each time-point, the average background signal per pixel was measured in the embryo cytoplasm, multiplied by the number of pixels in the segmented chromosomes, and this integrated background signal was then subtracted from the GFP signal on chromosomes. These values were then normalized at every timepoint by the average background value. Measurements were then aligned in time relative to NEBD detected in the DIC channel. For the HCP-3HFD::CPAR-1Ntail::GFP, the background signal was measured around the chromosomes in order to correct the measured signal for unspecific GFP localized in the nucleus prior to NEBD.

Kinetochore localizations of mCherry tagged BUB-1 mutants, mCherry tagged KNL-1 mutants, and of other proteins in these genetic backgrounds, were quantified at specific timepoints by proceeding to 10-pixel wide linescans in the spindle pole axis, and extracting the average of the signal extending 0.85 µm either side of the average chromosome position. This value was then normalized by the average background measured in the cytoplasm.

## Chromosome stretch quantification

Chromosome stretch quantification was carried out manually on the same movies as used for chromosome span quantification. Measurements were made on the maximal projection of the GFP channel, 20 s after anaphase onset detected as the time at which the chromosome span is seen starting to increase. Measurements were obtained from the average signal of a 3-pixel wide line, traced perpendicular to the spindle axis, between the segregating chromosome masses. This signal was normalized by the average signal obtained from an equivalent line measured between poles and chromosomes.

## Immunofluorescence and imaging

A Nikon CFI PLAN APO LBDA 100x/NA1.45 oil objective and a CoolSNAP HQ2 CCD camera (Photometrics Scientific) were used to aquire images of stained embryos with a z-step of 0.1 or 0.2 µm. Embryos were prepared following the protocol described in (*Gigant et al., 2017*). Custom-made antibodies against BUB-1, HCP-1, HCP-2 and KNL-1 were coupled to either dylight550 or dylight650 (Thermo Scientific) and were used each at a concentration of 1 µg/mL. An anti-α-tubulin diluted 1:100 (DM1α, Abcam) was used to stain microtubules, and chromosomes were stained with Hoechst at 2 µg/mL. Images were deconvolved using the Huygens software (Scientific Volume Imaging). Maximum projections of informative z-plan are presented.

## Embryonic lethality assays

Embryonic lethality was assayed after injecting L4 larvae with dsRNA, leaving them at 20°C during 48 hr for protein depletion, singling them out to lay embryos on plates placed at the specific temperatures of 15°C, 20°C or 24°C during 12 hr, before removing the parent worm and leaving the

progeny to develop for 24 hr. Embryonic lethality was then scored as the percentage of dead embryos found within the progeny.

## Graphs and statistical analysis

GraphPad Prism 6 (GraphPad Software) was used to generate graphs and proceed to statistical analysis. The tests used are mentioned in the figure legends.

## Acknowledgements

We thank all members of the Pintard and Dumont labs for support and advice. We thank Déborah Bourc'his for critical reading and editing of the manuscript. We are grateful to Patricia Moussounda, Soilihi Madi Ali and Christie Ouaddi for providing technical support. We thank Arshad Desai, Reto Gassmann and Dhanya Cheerambathur for worm strains. Some strains were provided by the CGC, which is funded by NIH Office of Research Infrastructure Programs (P40 OD010440). FE was supported by a PhD fellowship from the French Ministry of Research and from the *Association pour la Recherche sur le Cancer* (ARC). This work was supported by CNRS and University Paris Diderot, by grant NIH-R01GM117407 to JCC, and by grants from the *Mairie de Paris* (Emergence) and the *Fondation pour la Recherche Médicale* (FRM DEQ20160334869) to JD.

## Additional information

### Funding

| Funder | Grant reference number | Author |
|---|---|---|
| Fondation ARC pour la Recherche sur le Cancer | Doctorant en 4e année de thèse | Frances Edwards |
| Ministère de l'Enseignement supérieur, de la Recherche et de l'Innovation | | Frances Edwards |
| National Institutes of Health | R01GM117407 | Julie C Canman |
| Fondation pour la Recherche Médicale | DEQ20160334869 | Julien Dumont |
| Mairie de Paris | Emergence | Julien Dumont |

The funders had no role in study design, data collection and interpretation, or the decision to submit the work for publication.

### Author contributions

Frances Edwards, Conceptualization, Formal analysis, Investigation, Writing—original draft, Writing—review and editing; Gilliane Maton, Julie C Canman, Conceptualization; Nelly Gareil, Resources, Methodology; Julien Dumont, Conceptualization, Funding acquisition, Writing—original draft, Writing—review and editing

### Author ORCIDs

Julie C Canman https://orcid.org/0000-0001-8135-2072
Julien Dumont https://orcid.org/0000-0001-5312-9770

### Decision letter and Author response

Decision letter https://doi.org/10.7554/eLife.40690.056
Author response https://doi.org/10.7554/eLife.40690.057

## Additional files

### Supplementary files

• Supplementary file 1. (**A**) List of worm strains used in this study. (**B**) List of templates and primers used to synthesize double stranded RNA. (**C**) Targeted regions for CRISPR-Cas9 mediated *hcp-1$^{CENP-F}$* and *hcp-2$^{CENP-F}$* deletion mutant generation. (**D**) Images and datasets presented in different figures and panels.
DOI: https://doi.org/10.7554/eLife.40690.053

• Transparent reporting form
DOI: https://doi.org/10.7554/eLife.40690.054

### Data availability

All data generated or analysed during this study are included in the manuscript and supporting files.

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
