## [Decision Letter]

Thank you for submitting your article "BUB-1 promotes chromosome biorientation by coordinating antagonistic activities at the kinetochore" for consideration by *eLife*. Your article has been reviewed by four peer reviewers, including Jon Pines as the Reviewing Editor and Reviewer #1, and the evaluation has been overseen by a Reviewing Editor and Anna Akhmanova as the Senior Editor. The following individuals involved in review of your submission have agreed to reveal their identity: Jakob Nilsson (Reviewer #3).

The reviewers have discussed the reviews with one another and the Reviewing Editor has drafted this decision to help you prepare a revised submission.

Summary:

In this study the authors have investigated the role of BUB1 in chromosome segregation using the C. Elegant system. The find that BUB1 has opposing roles in microtubule attachment through interacting with the CENP-F/CLASP proteins and the RZZ/dynein-dynactin pathway. They provide evidence that BUB1 acts to co-ordinate these pathways by promoting microtubule attachment to kinetochores but preventing maturation of attachments by the SKA complex.

This is a careful study in which the authors have exploited the RNAi, imaging and genetic strengths of the *C. elegans* system to the full. The authors have tested a number of hypotheses to explain the function of BUB1 and present convincing evidence that the BUB1 does interact with two pathways influencing kinetochore-microtubule attachments (CENP-F/CLASP and RZZ/Dunedin/dynactin/SKA).

Essential revisions:

1) The study is very complicated for non-geneticists and very hard to follow. To make this study accessible to a large audience, the authors will need to re-write it and present the models/hypothesis on which their experiments are built up in a much clearer manner. For example: is it really appropriate to call BUB-1's functions "antagonistic" with regards to chromosome biorientation? Similarly, what is the basis for the statement that the BUB-1^ΔKD^ deletion mutant "is capable of limiting attachment maturation"? The BUB-1^ΔKD^ deletion is shown to suppresses sister chromatid co-segregation in the absence of HCP-1/2, so wouldn't this mutant therefore be predicted to be INCAPABLE of limiting attachment maturation, similar to the BUB-1^K718R; D847N^ mutant or BUB-1 depletion? The BUB-1^ΔKD^ deletion is informative for the analysis in Figure 6 because it delocalizes CLS-2 from kinetochores without causing the severe sister co-segregation phenotype associated with CLS-2 depletion. Is that what the authors mean?

2) The authors use only a few embryos (sometimes less than 10), and they re-use the same dataset several times. (For example: the four conditions used in Figure 1F are exactly the same cells used in Figure 4E, and 3 conditions are used again in Figure 5E.) The authors should state the number of independent experiments indicated and explain how they made sure that results are not due to day to day variability (due e.g. to small variations in temperature). When analyzing double mutants, the authors should analyse at least 3 single mutant embryos to make sure that the phenotype is robust. The authors should also point out when the data shown in different figures is the same dataset.

3) The authors should directly demonstrate one of the central assumptions of the study – that depletion of Bub1 leads to a large number of merotelic attachments – by staining for merotelic attachments. The images shown in Figure 1E do not allow one to distinguish between merotely and other type of segregation errors. The kymographs of kinetochores and centrosome markers may be an over-simplification as different errors of attachment could give similar kymographs.

4) Another central conclusion is that the presence of BUB1 at kinetochores antagonizes the binding of the SKA complex to kinetochores. However, only at the very end of the discussion do the authors point out that this role appears only in mutants lacking CLASP or CENP-F. Under physiological conditions, the presence or absence of BUB1 does not seem to influence SKA1 localization. In fact, the authors think that BUB1 only affects SKA1 because it is present at non-physiological levels in embryos lacking CLASP or CENP-F. It is therefore difficult to conclude that the physiological role of Bub1 is to antagonize the SKA complex. Under physiological conditions, this appears not to be the case, which strongly reduces the significance of their conclusions.

5) The authors should discuss the discrepancy between their conclusion in this study that BUB1 is required for the recruitment of the RZZ complex and the conclusions of Essex et al., 2009, that the RZZ complex is recruited to kinetochores independently of Bub1, and their own paper (Maton et al., 2015) that considers BUB1 and RZZ as part of separate branches.

6) The authors should address the apparent discrepancy that they assume that all the functions of CLASP and HCP1/2 originate from their role at kinetochores via BUB1, but in BUB-1 RNAi + BUB1^ΔKD^ embryos, HCP1/2/CENP-F are not present at kinetochores and chromosome segregation is normal (Figure 3—figure supplement 1). Nevertheless, when HCP1/2 are depleted, merotelic chromosomes appear. This would suggest that HCP1/2 do not require to be brought to kinetochores by BUB1 to prevent merotelic attachments.

---

## [Author Response]

Summary:In this study the authors have investigated the role of BUB1 in chromosome segregation using the C. Elegant system. The find that BUB1 has opposing roles in microtubule attachment through interacting with the CENP-F/CLASP proteins and the RZZ/dynein-dynactin pathway. They provide evidence that BUB1 acts to co-ordinate these pathways by promoting microtubule attachment to kinetochores but preventing maturation of attachments by the SKA complex.This is a careful study in which the authors have exploited the RNAi, imaging and genetic strengths of the C. elegans system to the full. The authors have tested a number of hypotheses to explain the function of BUB1 and present convincing evidence that the BUB1 does interact with two pathways influencing kinetochore-microtubule attachments (CENP-F/CLASP and RZZ/Dunedin/dynactin/SKA).

We are pleased the reviewers and editors were enthusiastic about our paper. We have performed additional experiments and made substantial changes to the text and figures to address all suggested revisions.

Essential revisions:1) The study is very complicated for non-geneticists and very hard to follow. To make this study accessible to a large audience, the authors will need to re-write it and present the models/hypothesis on which their experiments are built up in a much clearer manner. For example: is it really appropriate to call BUB-1's functions "antagonistic" with regards to chromosome biorientation?

We apologize for not being clearer in our original manuscript. Our revised manuscript has been drastically edited and re-written to enhance clarity of our main message. We have also removed some results (HIM-10 experiment on original Figure 4 and BUB-1 full depletion in the Δska-1 strain on original Figure 5), which were not essential for understanding the main message of our study and were introducing unnecessary complexity. We hope this will help the large audience of *eLife*, including non-geneticists, to follow our study.

We also agree that the use of the term ‘antagonistic’ to describe the various functions of BUB-1 was not appropriate. This term was in fact referring to the opposing effects of BUB-1 on chromosome biorientation: (1) HCP-1/2^CENP-F^ and CLS2^CLASP^-mediated promotion of biorientation and (2) prevention of biorientation through inhibition of SKA complex kinetochore recruitment. As this was potentially misleading, we edited our manuscript to better reflect BUB-1 key role in chromosome biorientation and segregation. We have for example edited the title of our manuscript to: “BUB-1 promotes amphitelic chromosome biorientation via multiple activities at the kinetochore”.

Similarly, what is the basis for the statement that the BUB-1^ΔKD^ deletion mutant "is capable of limiting attachment maturation"? The BUB-1^ΔKD^ deletion is shown to suppresses sister chromatid co-segregation in the absence of HCP-1/2, so wouldn't this mutant therefore be predicted to be INCAPABLE of limiting attachment maturation, similar to the BUB-1^K718R; D84^ mutant or BUB-1 depletion?

We thank the reviewers and editors for pointing this mistake to our attention and we apologize for this writing error. It should of course have been “is incapable of limiting attachment maturation”. This mistake has been edited in the revised version of our manuscript.

The BUB-1^ΔKD^ deletion is informative for the analysis in Figure 6 because it delocalizes CLS-2 from kinetochores without causing the severe sister co-segregation phenotype associated with CLS-2 depletion. Is that what the authors mean?

The reviewers and editors are absolutely right. This is exactly what we mean and the reason why we used this mutant in Figure 6. We have made this clearer in our revised manuscript.

2) The authors use only a few embryos (sometimes less than 10), and they re-use the same dataset several times. (For example: the four conditions used in Figure 1F are exactly the same cells used in Figure 4E, and 3 conditions are used again in Figure 5E.) The authors should state the number of independent experiments indicated and explain how they made sure that results are not due to day to day variability (due e.g. to small variations in temperature). When analyzing double mutants, the authors should analyse at least 3 single mutant embryos to make sure that the phenotype is robust. The authors should also point out when the data shown in different figures is the same dataset.

The high reproducibility of events in the *C. elegans* zygote does not require a particularly high number of individual embryos to be analyzed. However, for each experiment presented in this study, the sample size is equal or higher to sample size observed in most published studies on *C. elegans*. Since worms are treated (RNAi, Filming, staining, etc.) individually, each embryo could in theory be considered as an individual biological replicate. The same datasets are presented on several panels (Figure 1F and Figure 4E, Figure 3D and Figure 5E) for clarity only, and used as reference points for analyzing new data. We could in theory remove the already presented conditions from these panels, but this would complicate analysis of new data, as it would require the reader to scan through different figures. For clarity and transparency, we have now included Supplementary file 4, which displays images and datasets presented in different figures and panels. Importantly, kymographs were generated from different representative embryos when the same condition is presented in different panels. Furthermore, the same embryo was only and purposely used in cases when, for a same condition, one panel represents a single timepoint, and another panel represents a temporal series.

3) The authors should directly demonstrate one of the central assumptions of the study – that depletion of Bub1 leads to a large number of merotelic attachments – by staining for merotelic attachments. The images shown in Figure 1E do not allow one to distinguish between merotely and other type of segregation errors. The kymographs of kinetochores and centrosome markers may be an over-simplification as different errors of attachment could give similar kymographs.

Kinetochores are holocentric in the *C. elegans* nematode and form along the entire length of sister chromatids. Microtubules therefore attach all along the length of these extended kinetochores and do not form kinetochore fibers, as typically observed in monocentric species. For this reason, cold-treatments that specifically lead to spindle but not kinetochore microtubule depolymerization, and which are commonly used to analyze the type of attachment in monocentric cells, are inefficient in *C. elegans*. Instead, we have now performed immunofluorescence staining of spindle microtubules and kinetochores to clarify the attachment status in various conditions, including in BUB-1-depleted *C. elegans* zygotes. Importantly, these results allowed the detection of bent kinetochores during metaphase in absence of BUB-1, unambiguously demonstrating the presence of a majority of merotelic attachments in this condition. These new results have been included in a revised Figure 1 (panel F).

4) Another central conclusion is that the presence of BUB1 at kinetochores antagonizes the binding of the SKA complex to kinetochores. However, only at the very end of the discussion do the authors point out that this role appears only in mutants lacking CLASP or CENP-F. Under physiological conditions, the presence or absence of BUB-1 does not seem to influence SKA1 localization. In fact, the authors think that BUB-1 only affects SKA1 because it is present at non-physiological levels in embryos lacking CLASP or CENP-F. It is therefore difficult to conclude that the physiological role of BUB-1 is to antagonize the SKA complex. Under physiological conditions, this appears not to be the case, which strongly reduces the significance of their conclusions.

We respectfully disagree with that assertion. We instead argue that HCP-1/2^CENP-F^ or CLS-2^CLASP^ depletions allow to ‘amplify’ and thus to highlight a physiological process that is otherwise less apparent and probably extremely transient at the beginning of mitosis. The embryonic lethality associated with a high degree of merotely observed in the BUB-1 mutants incapable of restricting biorientation further demonstrates the essential and physiological role of this pathway (Figure 6A-C). In the same way that outer kinetochore expansion (which physiologically occurs transiently during prometaphase in tissue cultured cells, when kinetochores are not yet attached to microtubules) is primarily observed and studied in response to drug-mediated depolymerization of microtubules, this novel physiological function of BUB-1 is highlighted in conditions that decrease kinetochore microtubule dynamics. We have now added this notion to our revised discussion for clarity.

5) The authors should discuss the discrepancy between their conclusion in this study that BUB1 is required for the recruitment of the RZZ complex and the conclusions of Essex et al., 2009, that the RZZ complex is recruited to kinetochores independently of BUB1, and their own paper (Maton et al., 2015) that considers BUB1 and RZZ as part of separate branches.

Our results are consistent with the role for BUB1 in recruiting the RZZ complex to kinetochores in human cells (Caldas et al., 2015; Zhang et al., 2015). They are based on a careful quantification of ZWL-1^Zwilch^::GFP at kinetochores over time in presence or absence of BUB-1 (Figure 4B and Figure 4—figure supplement 1A,B). In contrast, the conclusion from Essex et al., 2009 was drawn from immunofluorescent staining in two-cell stage embryos with activated SAC through monopolar spindles (Figure 4G) and this staining was not specifically quantified. Furthermore, a significant reduction and premature disappearance of kinetochore-localized GFP::CZW-1^ZW10^ (another member of the RZZ complex) was in fact visible in absence of BUB-1 at kinetochores in our previous work on the central spindle (Supplementary Figure 3C, Maton et al., 2015). At the time, we did not focus on this phenotype associated with the KNL-1 Δ85-505 mutant, as the effect of this mutant on HCP-1/2^CENP-F^ and CLS-2^CLASP^ kinetochore localizations was a lot more potent. However, in light of our new results presented in the current manuscript, these previous data are fully consistent with a role for kinetochore-localized BUB-1 in the recruitment of the RZZ complex. We have thus edited our revised manuscript to cite this previous work where appropriate.

6) The authors should address the apparent discrepancy that they assume that all the functions of CLASP and HCP1/2 originate from their role at kinetochores via BUB1, but in BUB-1 RNAi + BUB1^ΔKD^ embryos, HCP1/2/CENP-F are not present at kinetochores and chromosome segregation is normal (Figure 3—figure supplement 1). Nevertheless, when HCP1/2 are depleted, merotelic chromosomes appear. This would suggest that HCP1/2 do not require to be brought to kinetochores by BUB1 to prevent merotelic attachments.

We thank the reviewers and editors for bringing this concern to our attention and we apologize for being unclear about this in the original version of our manuscript. We in fact do not assume that all functions of CLASP and HCP-1/2 are at the kinetochore, and we even have evidence (as suggested by the Reviewers and Editors: the higher rate of merotely in absence of HCP-1/2 in the BUB1^Δ*KD*^ mutant) against this premise. Our results are consistent with the idea that CLASP and HCP-1/2 have roles both at and outside the kinetochore. As our current study does not focus precisely on the function of CLASP and HCP-1/2, we did not elaborate on that specific point, but we have tried to clarify this in the revised version of our manuscript.